# Continuity-Regularized Flow Matching for Offline Reinforcement Learning

Xiaocong Chen [1]   Siyu Wang [2]   Lina Yao [3]

## Abstract

Flow-matching policies have recently emerged as a powerful class of generative models for offline reinforcement learning (RL), capable of capturing complex, multi-modal action distributions from static datasets. However, standard training objectives are largely agnostic to the global properties of the generative path, permitting learned vector fields that are irregular and unstable, which can hinder performance. In this work, we introduce PDE-regularized Q-Learning (PQL), a novel algorithm that addresses this limitation by imposing a principled structure on the entire probability flow. PQL makes two synergistic contributions: first, a partial differential equation based regularizer derived from the continuity equation promotes global smoothness and stability on the flow. Second, to solve the complex optimization problem introduced by this regularizer, we propose a Beta-distributed timestep sampling strategy that focuses learning on the critical trajectory segments where the trade-off between imitation and smoothness is most acute. Through extensive experiments, we demonstrate that by structuring the generative journey and not just its destination, PQL achieves state-of-the-art performance on a wide range of challenging offline RL tasks.

## 1. Introduction

Recent advances in offline reinforcement learning (RL) have been driven by expressive generative policies capable of learning from static datasets with complex, multimodal action distributions (Wang et al., 2023; Zhu et al., 2023). Building on this progress, flow matching (FM) (Lipman et al., 2023) has emerged as a compelling alternative to diffusion models, delivering state-of-the-art performance on several

offline RL benchmarks while offering a simpler, more efficient training paradigm (Park et al., 2025b; Zheng et al., 2023; Zhang et al., 2025). FM learns a time-dependent vector field defining an ordinary differential equation (ODE) that deterministically transports samples from a simple prior to a target action distribution, thus sidestepping the costly iterative sampling of diffusion-based approaches (Lipman et al., 2024).

Despite their effectiveness, flow-matching policies for offline RL face a subtle but consequential failure mode that arises specifically from value-guided policy improvement. For a pure imitation flow, the continuity equation is satisfied by construction: any sufficiently regular velocity field $u_\theta(s, a, t)$ induces a density path that automatically conserves probability mass (Santambrogio, 2015). The standard flow-matching loss is pointwise (Lipman et al., 2023; Dao et al., 2023; Liu, 2022), but as long as the actor is trained only to imitate the dataset, this is not a problem—the resulting transport map is well-behaved. The difficulty arises when the actor is also trained to maximize an offline critic $Q_\phi$: the critic gradient pushes $u_\theta$ toward actions with high estimated value, which often lie near or beyond the support of the dataset. This Q-pressure actively deforms the transport map, producing locally large Jacobians under which nearby latent samples are mapped to distant action regions where the critic is unreliable. The deformation is invisible to a pointwise flow-matching objective, yet directly degrades the policy-improvement signal that downstream components, such as distilled one-step actors (Park et al., 2025b), rely on.

We address this RL-specific deformation by regularizing the geometry of the critic-guided transport map. The continuity equation (Risken, 1989) is not a constraint to be enforced from outside; rather, it is the analytical framework that links density evolution to local Jacobian properties of $u_\theta$. Standard transport-theoretic results show that bounding the Jacobian norm of the velocity field controls both its Lipschitz constant and the divergence governing density spread. Inspired by neural-ODE regularization in generative modeling (Finlay et al., 2020), we therefore penalize the Frobenius norm of the Jacobian $\nabla_a u_\theta$ as a tractable surrogate for transport regularity. Our novelty is not the regularizer itself but (i) identifying that critic-guided policy improvement is the source of flow irregularity in flow-based offline RL, and

[1]CSIRO's Data 61, Australia [2]School of Computing, Macquarie University, Australia [3]School of Computer Science and Engineering, University of New South Wales, Australia. Correspondence to: Xiaocong Chen <xiaocong.chen@data61.csiro.au>.

*Proceedings of the 43rd International Conference on Machine Learning*, Seoul, South Korea. PMLR 306, 2026. Copyright 2026 by the author(s).

(ii) showing that effectively integrating this regularizer in the RL setting requires a non-uniform timestep sampling scheme.

However, incorporating this PDE constraint introduces a new optimization challenge. The model must now learn a vector field that both matches the target data flow and satisfies the geometric constraints imposed by the regularizer. We argue that the standard practice of sampling time $t$ uniformly is not suited for this more complicated landscape. This is because the regularizer's influence is strongest not at the endpoints, but in the critical mid-range of the trajectory where an unconstrained model would learn its most complex and irregular dynamics (Karras et al., 2022; Kingma et al., 2021). Uniform sampling under-allocates attention to these regions where the trade-off between imitation accuracy and transport regularity is most acute. To resolve this, we propose Beta-distributed time sampling, which focuses updates on this challenging regime to find a stable and effective solution for the PDE-constrained objective. Our contributions are threefold:

- We introduce a Jacobian-based regularizer, motivated by the continuity equation, that stabilizes the critic-guided transport map in flow-based offline RL, mitigating the policy-quality degradation caused by Q-guided deformation of the flow.

- We design a Beta-distributed time sampling strategy that enables stable and efficient optimization of our PDE-constrained objective by focusing updates on the most challenging temporal regions of the flow.

- We demonstrate empirically that our method, which we call PQL, achieves state-of-the-art performance and significantly improves training stability on standard offline RL benchmarks.

**Conflict of Interest Disclosure.** The authors declare that they have no financial conflicts of interest related to this work.

## 2. Related Work

**Diffusion Models for Offline RL.** Diffusion models (Hansen-Estruch et al., 2023; Ding et al., 2024; Chen et al., 2024; Ding & Jin, 2024) have recently been adapted with great success to offline RL, offering a powerful tool for learning complex policies. Frameworks like Diffuser (Janner et al., 2022) treat decision-making as a trajectory denoising problem, generating entire plans that are then guided toward high-return outcomes. In robotics, diffusion-based planning has shown performance comparable to traditional methods while better capturing multi-modal behaviors (Chi et al., 2023; Chen et al., 2025). Other approaches treat policies as

conditional diffusion processes, iteratively refining actions from noise, which provides stable training and naturally handles multi-modal action distributions in high-dimensional spaces (Wang et al., 2023). Subsequent work has improved both the guidance and efficiency of diffusion policies: Lu et al. (2023) introduce contrastive energy prediction as an exact, training-time alternative to Q-weighted guidance, and Kang et al. (2023) propose distillation-based diffusion policies that substantially reduce sampling cost while remaining compatible with a range of offline RL objectives.

**Flow Matching for Offline RL.** Flow Matching (FM) has emerged as a highly efficient and deterministic alternative to diffusion for generative policy learning. FM models learn a vector field that defines an Ordinary Differential Equation (ODE) to transport samples from a simple prior to the target action distribution. This approach has demonstrated superior performance in continuous control tasks by leveraging the stability and efficiency of ODE-based generation (Lipman et al., 2023; 2024). In offline RL, FM has been applied to learn latent action spaces that ensure conservatism (Akimov et al., 2022), to perform value-based guidance by weighting the flow objective with energy functions (Zhang et al., 2025), and to incorporate guidance directly into the training of the velocity field (Alles et al., 2025). Flow-based methods are well-suited to offline RL because their continuous, deterministic transport paths admit principled regularization and efficient sampling, while still capturing the complex, multi-modal action distributions present in static datasets (Park et al., 2025b). Closer to the critic side, Agrawalla et al. (2026) concurrently parameterize the Q-function itself as a flow-matching model; their work is complementary to ours, since they regularize the value representation while we regularize the actor's transport map.

## 3. Preliminaries

### 3.1. Generative Flow Matching

Flow matching (Liu et al., 2023) is a technique for training continuous-time generative models. The goal is to learn a parameterized vector field $u_\theta(x, t)$ that defines a probability flow $\{p_t\}_{t \in [0,1]}$ capable of transforming a simple prior distribution $p_0$ (e.g., a standard Gaussian $\mathcal{N}(0, I)$) into a complex data distribution $p_1$. The evolution of a sample $x$ over this flow is described by the ordinary differential equation (ODE):

$$\frac{dx_t}{dt} = u(x_t, t), \quad \text{with} \quad x_0 \sim p_0. \quad (1)$$

Generating a sample $x_1 \sim p_1$ is achieved by integrating this ODE from $t = 0$ to $t = 1$. The challenge lies in defining a suitable target vector field $u(x, t)$ for training.

Conditional Flow Matching (CFM) provides an elegant solution by defining a simple, fixed path between any pair of samples $x_0 \sim p_0$ and $x_1 \sim p_1$. A common choice is the linear interpolation path $x_t = (1-t)x_0 + tx_1$. The target vector field along this path is simply its time derivative, which is constant: $v \triangleq \frac{dx_t}{dt} = x_1 - x_0$. The model $u_\theta(x_t, t)$ is then trained to regress onto this target field $v$ for all points $(x_t, t)$ along the path. This yields the flow matching loss:

$$\mathcal{L}_{\text{FM}}(\theta) = \mathbb{E}_{x_0 \sim p_0, x_1 \sim p_1, t \sim \mathcal{U}(0,1)}\big[\|u_\theta(x_t, t) - (x_1 - x_0)\|^2\big]. \quad (2)$$

This objective allows for stable and efficient training of generative models without requiring simulations or adversarial training.

### 3.2. Offline Reinforcement Learning with Flow Matching

In offline reinforcement learning, the goal is to learn a high-performing policy from a static dataset $\mathcal{D} = \{(s, a, r, s')\}$ collected by one or more unknown behavior policies. We can frame policy learning as a generative modeling problem: we want to learn a conditional generator $\pi(a|s)$ that produces high-value actions.

Flow matching is a natural fit for this task. We learn a conditional vector field $u_\theta(s, a, t)$ that transforms a noise vector $a_0 \sim \mathcal{N}(0, I)$ into a desirable action $a_1$. To begin, we can train the model to simply imitate the actions in the dataset. This corresponds to a conditional flow matching objective where the target actions $a_1$ are drawn from the dataset $\mathcal{D}$:

$$\mathcal{L}_{\text{imitate}}(\theta) =$$
$$\mathbb{E}_{(s,a_1) \sim \mathcal{D}, a_0 \sim \mathcal{N}(0,I), t \sim \mathcal{U}(0,1)}\big[\|u_\theta(s, a_t, t) - (a_1 - a_0)\|^2\big], \quad (3)$$

where $a_t = (1-t)a_0 + ta_1$. This objective trains the policy to reproduce the action distribution $p_\mathcal{D}(a|s)$ found in the data. However, since the offline dataset is often suboptimal, pure imitation is insufficient. Modern approaches therefore augment this objective with value-based or reward-weighted terms to bias the learned policy toward actions that are expected to yield higher returns than those in the dataset. This framing provides the foundation for our proposed algorithm.

## 4. Methodology

### 4.1. PDE-Guided Regularized Policy Flows

As established in our preliminaries, we can train a conditional flow model to imitate the actions in an offline dataset using the objective $\mathcal{L}_{\text{imitate}}$ (Equation (3)). Building upon recent advancements in flow-based reinforcement learning (Park et al., 2025b), we move beyond simple imitation by combining the imitation loss with a policy improvement term that encourages the flow to generate actions with high

estimated Q-values:

$$\mathcal{L}_{\text{imitate}}(\theta) - \mathbb{E}_{s \sim \mathcal{D}, a_0 \sim \mathcal{N}(0,I)}\big[Q_\phi(s, a_1')\big], \quad (4)$$

where $a_1'$ is the action produced by integrating the learned vector field $u_\theta$ from $a_0$ over $t \in [0, 1]$. Throughout this section, expectations involving $(s, a_0, a_1)$ are taken with respect to $s \sim \mathcal{D}$, $a_0 \sim \mathcal{N}(0, I)$, and $a_1 \sim p_\mathcal{D}(\cdot \mid s)$, with the interpolant $a_t = (1-t)a_0 + t a_1$; the distribution over $t$ is stated explicitly in each expression.

**Why regularize? Critic-induced flow deformation.** A pure imitation flow ($\mathcal{L}_{\text{imitate}}$ alone) automatically satisfies the continuity equation,

$$\partial_t \rho_t(a \mid s) + \nabla_a \cdot \big(\rho_t(a \mid s)\, u_\theta(s, a, t)\big) = 0, \quad (5)$$

ensuring that probability mass is conserved along the trajectories of $u_\theta$ (Santambrogio, 2015). Continuity is therefore not a constraint we must impose externally; it is a structural property of any sufficiently regular velocity field. The difficulty in offline RL is that the critic term $-\mathbb{E}[Q_\phi(s, a_1')]$ in Equation (4) does not preserve this structure. Because $Q_\phi$ is reliable only within the support of $\mathcal{D}$, gradients from this term push $u_\theta$ toward weakly-supported, high-value regions, deforming the transport map. The deformation manifests as locally large Jacobians of $u_\theta$, where nearby latent samples are mapped to distant action regions and the critic's errors are amplified rather than averaged out. We therefore do not aim to enforce Equation (5) as a residual; we instead control the local geometry of $u_\theta$ so that the critic-guided transport map remains well-behaved.

**From continuity to Jacobian control.** Standard transport-theoretic results (see Theorem 4.1) show that the regularity of the density path $\rho_t(\cdot|s)$ induced by $u_\theta$ is governed by the Lipschitz constant and divergence of $u_\theta$ in $a$, both of which are bounded by the Frobenius norm of the Jacobian $\nabla_a u_\theta$. Following neural-ODE regularization in generative modeling (Finlay et al., 2020), we therefore introduce a Jacobian-based regularizer:

$$\mathcal{L}_{\text{PDE}}(\theta) = \lambda\, \mathbb{E}_{s, a_0, a_1,\, t \sim \mathcal{U}(0,1)}\big[\|\nabla_a u_\theta(s, a_t, t)\|_F^2\big]. \quad (6)$$

Penalizing the Frobenius norm of the Jacobian directly limits all local deformations of the flow—including shear and rotation in addition to expansion and compression. Unlike a simpler divergence-only penalty, this constrains the full local geometry of the flow, which we find is essential for keeping critic-guided transport stable and for preventing the collapse of the action distribution during training. We emphasize that, while the technique of Jacobian regularization is borrowed from Finlay et al. (2020), who use it to reduce numerical integration cost in neural ODEs, in our setting it serves a different purpose: it counteracts an actively destabilizing optimization pressure produced by the critic, which is unique to the value-guided RL setting.

**Theoretical Guarantees.** Our Jacobian-based regularizer is motivated by its ability to formally stabilize the learned probability flow relative to a clean reference path. We take this reference to be the conditional interpolation flow underlying Equation (3): namely, $u_\star(a, t) = a_1 - a_0$ with $a_0 \sim \mathcal{N}(0, I)$ and $a_1 \sim p_{\mathcal{D}}(\cdot \mid s)$, and let $\rho_t^\star(\cdot \mid s)$ denote the marginal density induced by this reference flow at time $t$. The following theorem bounds the Wasserstein distance between the learned density path $\rho_t^\theta$ and this reference $\rho_t^\star$, explicitly in terms of the Jacobian norm that our method controls.

**Theorem 4.1** (Path Stability via Jacobian Control). *Let $u_\theta$ and $u_\star$ be two velocity fields whose Jacobians in $a$ are both uniformly bounded by $J$, i.e., $\|\nabla_a u_\theta(s, a, t)\|_F \leq J$ and $\|\nabla_a u_\star(s, a, t)\|_F \leq J$ for all $(s, a, t)$. Then for any $t \in [0, 1]$, the corresponding density paths $\rho_t^\theta$ and $\rho_t^\star$ satisfy*

$$
W_2\big(\rho_t^\theta(\cdot \mid s), \, \rho_t^\star(\cdot \mid s)\big)
$$
$$
\leq \ e^{(1+\sqrt{d}) J t} \cdot \int_0^t \Big( \mathbb{E}_{a \sim \rho_\tau^\star(\cdot \mid s)} \big[ \Delta_{\theta, \star} \big] \Big)^{1/2} d\tau, \quad (7)
$$

*where $\Delta_{\theta, \star}(s, a, \tau) = \|u_\theta(s, a, \tau) - u_\star(s, a, \tau)\|^2$ and $d$ is the action dimension.*

A brief proof is provided in Section B. The key insight is that bounding the Jacobian's Frobenius norm by $J$ provides an upper bound on both the Lipschitz constant ($L \leq J$) and the magnitude of the divergence ($|\nabla_a \cdot u_\theta| \leq \sqrt{d} J$). This theorem shows that by minimizing our regularizer $\mathcal{L}_{\text{PDE}}$ (Equation (6)), we directly tighten the stability bound on the learned policy flow relative to the reference interpolation flow. We emphasize, however, the scope of this guarantee: Theorem 4.1 motivates Jacobian control of the transport map and bounds how far the learned density path can drift from the reference, but it does not by itself guarantee return optimality, which is determined by the critic and the policy-improvement objective rather than by the geometry of the BC-flow alone. We note that Theorem 4.1 requires only a uniform bound on the Frobenius norm of $\nabla_a u_\theta$; no separate global Lipschitz assumption on $u_\theta$ is needed, since the operator norm of the Jacobian (which equals its Lipschitz constant in $a$) is always upper-bounded by the Frobenius norm. Our regularizer $\mathcal{L}_{\text{PDE}}$ controls this Frobenius bound directly during training.

The exponential dependence on $J$ in the bound of Theorem 4.1 is not specific to our method; it arises from standard applications of Grönwall's inequality in the stability analysis of ordinary differential equations. In practice, our flow time is restricted to $t \in [0, 1]$, so the factor $e^{(1+\sqrt{d}) J t}$ remains well behaved as long as $J$ is kept moderate. Without any regularization, the effective Jacobian norm can grow large and the bound becomes vacuous, which matches the unstable paths we observe empirically. By contrast, our Jacobian regularizer actively drives $J$ down during training,

which tightens the bound and yields more stable generative paths.

To apply this regularization, the regularizer in Equation (6) is estimated efficiently without materializing the full Jacobian matrix by using Hutchinson's trace estimator. This method relies on Jacobian-vector products (JVPs), where the Jacobian is multiplied by a random probe vector $z$. We define $z$ as a vector sampled from a standard multivariate normal distribution, i.e., $z \sim \mathcal{N}(0, I)$, where $I$ is the identity matrix of the same dimension as the action space.

The JVP is the directional derivative of the vector field $u_\theta$ in the direction of $z$: $\text{JVP}(u_\theta; z) \triangleq \nabla_a u_\theta(s, a, t) \cdot z$. Modern deep learning frameworks compute this efficiently using forward-mode automatic differentiation. The squared Frobenius norm of the Jacobian is then recovered through the stochastic identity:

$$
\|\nabla_a u_\theta(s, a, t)\|_F^2 \ = \ \mathbb{E}_{z \sim \mathcal{N}(0, I)} \|\text{JVP}(u_\theta; z)\|^2. \quad (8)
$$

In practice, a low-variance estimate is obtained using just one or two random probe vectors per sample, making the method scalable to high-dimensional action spaces.

**Regularized Actor Objective.** We now assemble the full actor loss by combining the conditional flow-matching objective with the critic-guided policy improvement term and our Jacobian regularizer $\mathcal{L}_{\text{PDE}}$. Using a learned critic $Q_\phi(s, a)$ to align the generative policy with the RL objective of maximizing returns, the final regularized actor objective is:

$$
\mathcal{L}_{\text{actor}}(\theta) = \mathbb{E}_{s, a_0, a_1, \, t \sim \mathcal{U}(0, 1)} \Big[ \big\| u_\theta(s, a_t, t) - (a_1 - a_0) \big\|^2 \Big]
$$
$$
- \mathbb{E}_{s, a_0} \big[ Q_\phi(s, a_1') \big] + \mathcal{L}_{\text{distill}} + \mathcal{L}_{\text{PDE}}(\theta), \quad (9)
$$

where $a_1'$ is the action generated by integrating the learned field $u_\theta$ from $a_0$ at $t = 0$ to $t = 1$, and $\mathcal{L}_{\text{distill}}$ is the distillation term from Park et al. (2025b) that encourages the one-step vector field prediction to align with this final generated action.

Although Equation (9) contains four terms, only the flow-matching loss and $\mathcal{L}_{\text{PDE}}$ depend on the continuous BC-flow $u_\theta(s, a, t)$ over $t \in [0, 1]$. The Jacobian regularizer is therefore imposed solely on this branch, since it is the branch that defines a full transport path and that is directly deformed by the critic gradient through the integrated action $a_1'$. The distilled one-step actor does not define such a time-dependent trajectory, so a continuity-based penalty would not be well-defined on it. This division is not a separation between generative modeling and policy learning: in offline RL the critic-guided flow is the source of the high-value targets that the deployed one-step policy distills from, so improving the geometry of the BC-flow directly changes

the policy-improvement signal the one-step actor receives. Our RL-specific claim is therefore not that Jacobian regularization is unique to RL, but that in flow-based offline RL the critic-guided transport branch is the locus where unsupported actions and unstable policy improvement arise—and this is precisely the branch we stabilize.

By minimizing the PDE term in this objective, we directly tighten the stability bound of Theorem 4.1. However, adding such a constraint without adapting the training process can be suboptimal (we demonstrate this in Section 5.4). By forcing the model to find a smoother solution, the regularizer makes the learning landscape more demanding. A naive uniform sampling strategy may not provide a strong enough signal for the model to find a high-performing policy within this newly constrained landscape. This motivates our second contribution: an adaptive sampling strategy designed to provide the focused learning signal necessary to unlock the full potential of the regularized flow.

### 4.2. Adaptive Timestep Sampling via Beta Distribution

As we just motivated, the introduction of the $\mathcal{L}_{\mathrm{PDE}}$ regularizer modifies the optimization landscape that the flow-matching objective is operating in. The model must now find a vector field $u_\theta$ that is both accurate to the target flow and geometrically simple, as measured by its Jacobian norm. This dual requirement presents a non-uniform challenge across the trajectory $t \in [0, 1]$. The regularizer's smoothing effect is most impactful in regions where an unconstrained model would otherwise learn a complex field with high curvature or divergence. We posit that these critical regions typically occur mid-trajectory, since the flow dynamics are constrained by simpler boundary conditions (pure noise and clean data) at the endpoints, leaving the most complex geometric transformations for the intermediary steps.

A standard uniform sampling strategy, which allocates an equal training budget to all timesteps, is poorly matched to this non-uniform challenge. To address this, we build from the key insight that the standard $Unif(0, 1)$ distribution is simply a special case of the Beta distribution: $Unif(0, 1) \equiv Beta(1, 1)$. This provides a principled and simple family of distributions to explore.

**Proposition 4.2** (Optimal Sampling for Variance Reduction). *Let $\mathcal{L}(\theta) = \int_0^1 \ell(t)\, dt$ be the total regularized loss. To minimize the variance of the stochastic gradient estimator for $\nabla_\theta \mathcal{L}$, the optimal sampling distribution $\pi^*(t)$ is proportional to the norm of the instantaneous gradient: $\pi^*(t) \propto \|\nabla_\theta \ell(t)\|$.*

Based on this principle, we introduce adaptive sampling specifically to the imitation component of our objective. We focus sampling on the imitation loss as it is the primary driver of the flow's path, while the $\mathcal{L}_{\mathrm{PDE}}$ term acts

as a global constraint on the vector field's geometry, which we found benefits from uniform sampling across the entire trajectory.

Therefore, we define an adaptively sampled imitation loss, $\mathcal{L}$, which replaces the uniform sampling in the first part of the $\mathcal{L}_{\mathrm{actor}}$ term:

$$\mathcal{L}(\theta) =$$
$$\mathbb{E}_{s,a_0,a_1,\, t\sim\mathrm{Beta}(\alpha,\alpha)} \left[ w_t^\pi \cdot \| u_\theta(s, a_t, t) - (a_1 - a_0) \|^2 \right]. \tag{10}$$

where $t \sim \mathrm{Beta}(\alpha, \alpha)$ with $\alpha > 1$, and $w_t^\pi = \frac{t}{1-t}\pi(t)$ with Beta distribution density $\pi(t)$. This strategy focuses the model's capacity on the segments of the flow where the trade-off between imitation accuracy and transport regularity is most critical, leading to a more stable and efficient learning process.

## 5. Experiment

In this section, we empirically evaluate the performance of PQL, comparing it to previous offline RL approaches on a variety of challenging tasks. We also provide extensive analyses and ablations on the effectiveness of different components and PQL's design choices. Extra experiments can be found in Section D.

### 5.1. Experimental Setup

We conduct a comprehensive evaluation of our proposed algorithm on the D4RL (Fu et al., 2020) and OGBench (Park et al., 2025a) benchmarks, following the standard experimental protocols established in prior work (Park et al., 2025b). To ensure a fair comparison across environments with different reward scales, we use the standard D4RL normalized score as our primary performance metric (Fu et al., 2020). Each method is trained for $10^6$ gradient steps (reduced to $5 \times 10^5$ for D4RL and pixel-based OGBench, following Park et al. (2025b)) and evaluated over 8 random seeds.

**Baselines.** We benchmark our algorithm against a comprehensive set of state-of-the-art methods spanning three major categories: traditional, diffusion-based, and flow-based offline RL. To ensure a fair and robust comparison, we select specific baselines for different task suites based on the availability of established implementations in the literature (Hu et al., 2025).

For the D4RL Gym-MuJoCo benchmarks, we compare against traditional methods including BC, IQL (Kostrikov et al., 2022), and CQL (Kumar et al., 2020); diffusion-based methods such as IDQL (Hansen-Estruch et al., 2023), SRPO (Chen et al., 2024), and CAC (Ding & Jin, 2024); and flow-based methods FQL (Park et al., 2025b), Flow (Zheng et al., 2023), and CNF (Akimov et al., 2022).

*Table 1.* Offline RL algorithms comparison on D4RL Gym-MuJoCo environments, grouped by task. Red indicates the best results while blue indicates the second best.

| Algorithm | HalfCheetah | | | Hopper | | | Walker2d | | |
|---|---|---|---|---|---|---|---|---|---|
| | Medium-Expert | Medium | Medium-Replay | Medium-Expert | Medium | Medium-Replay | Medium-Expert | Medium | Medium-Replay |
| BC | 55.2 | 42.6 | 36.6 | 52.5 | 52.9 | 18.1 | 107.5 | 75.3 | 26.0 |
| IQL | 93±3 | 50±0.2 | 42±4 | 86±30 | 65±4 | 90±13 | 112±0.5 | 81±3 | 75±9 |
| CQL | 62±3 | 45±2 | 47±2 | 99±5 | 58±3 | 49±2 | 110±5 | 79±4 | 27±2 |
| IDQL | 94±3 | 50±1 | 45±0.8 | 105±3 | 63±2 | 82±10 | 112±0.7 | 71±10 | 82±3 |
| SRPO | 92±3 | 60±0.8 | 51±3 | 100±14 | 95±2 | 101±1 | 114±2 | 84±4 | 84±7 |
| CAC | 59±4 | 69±0.7 | 59±4 | 100±4 | 81±11 | 99±0.5 | 110±0.7 | 81±10 | 79±4 |
| FQL | 86.1±6 | 60±3 | 53±6 | 21±3 | 25±0.8 | 28±3 | 13±10 | 9±2 | 7±3 |
| Flow | 97±1 | 49±2 | 42±0.8 | 105±3 | 84±2 | 89±6 | 94±1 | 77±2 | 78±0.9 |
| CNF | 96±0.3 | 51±0.5 | 46±0.2 | 109±5 | 69.3±1 | 89±10 | 112±0.5 | 84±3 | 82±2 |
| PQL(Ours) | 90±1 | 54±2 | 51±0.3 | 112±3 | 84±1 | 92±3 | 114±0.8 | 86±3 | 84±1 |

For the more complex Adroit and OGBench manipulation tasks, our comparison suite includes a different set of established baselines, with the flow-based methods comprising FAWAC and FBRAC (Park et al., 2025b).

## 5.2. Offline Evaluation

Our proposed algorithm, PQL, establishes a new state-of-the-art on the D4RL Gym-MuJoCo benchmarks, as detailed in Table 1. The gains are strongest when the dataset contains high-quality trajectories: PQL attains the top or tied-for-top score in five of the six Medium-Expert and Medium-Replay settings. Compared to leading baselines, PQL consistently improves over prior flow-based methods and outperforms strong diffusion-based models such as SRPO and CNF on the hardest Medium-Expert tasks. PQL also remains competitive on noisier Medium datasets, achieving the best score on Walker2d and strong performance on Hopper, which indicates that our regularization and sampling choices do not rely on near-expert data.

On the Adroit manipulation tasks, PQL shows a clear advantage over prior methods, as shown in Table 2. It achieves the best result in 11 out of the 12 task settings and is especially strong on cloned and expert datasets, where it obtains the highest scores across all four environments. PQL also performs well on the human datasets despite their noise and suboptimality, including the second-best score on pen-human and strong results on hammer-human relative to most diffusion and flow-based baselines. Overall, these results suggest that the PDE regularizer and Beta sampling help extract and stabilize skills from high-quality demonstrations while remaining robust under imperfect data.

Our analysis also includes the OGBench suite, which tests generalization across navigation and manipulation tasks. As demonstrated in Table 3, PQL achieves the top score in 8

out of the 10 evaluation settings. The largest gains appear in long-horizon navigation tasks such as Antmaze and Humanoid, where PQL substantially outperforms prior methods, and in dexterous manipulation tasks in Cube, Scene, and Puzzle, where it exceeds all baselines by a large margin. Taken together, the results across D4RL, Adroit, and OGBench support our core hypothesis: controlling the geometry of the critic-guided transport map, not only the terminal action distribution, leads to stronger offline RL policies across both high-quality and noisy datasets.

## 5.3. Design Choice

**Jacobian Regularization vs. Divergence Penalty**  A core contribution of our work is the PDE-guided regularizer, which penalizes the squared Frobenius norm of the Jacobian ($\|\nabla_a u_\theta\|_F^2$). A simpler alternative would be to penalize only the divergence of the vector field ($\nabla_a \cdot u_\theta$) as suggested in prior work (Huang et al., 2025). Our choice to penalize the full Jacobian is deliberate, as it provides a more comprehensive and robust form of regularization critical for policy stability by constraining not just the expansion and compression of the flow, but also its rotational and shear dynamics.

To empirically validate this choice, we conduct an experiment comparing our full PQL algorithm against a variant using a divergence-only penalty, denoted "PQL with Div". The learning curves across the nine D4RL Gym-MuJoCo environments are presented in Figure 1. While the divergence-only regularizer offers some benefit, the full Jacobian penalty is the more reliable choice across the suite. The gap is modest in some settings (both variants reach comparable performance on several Medium tasks), but the clearer advantages emerge on the harder "Medium-Expert" and "Medium-Replay" datasets. For example, on "Hopper-MR" and "Walker2d-ME", the Jacobian variant maintains

*Table 2.* Algorithm comparison across Adroit environments, grouped by task. Red indicates the best results while blue indicates the second best.

| | Pen | | | Door | | | Hammer | | | Relocate | | |
|---|---|---|---|---|---|---|---|---|---|---|---|---|
| Algorithm | Human | Cloned | Expert | Human | Cloned | Expert | Human | Cloned | Expert | Human | Cloned | Expert |
| BC | 34 | 57 | 85 | 0.5 | -0.1 | 35 | 2 | 0.8 | 126 | 0.0 | -0.1 | 101 |
| IQL | 82±18 | 77±18 | 134±16 | 3±2 | 0.8±1 | 105±3 | 3±2 | 1±0.5 | 130±0.5 | 0.1±0.1 | 0.2±0.4 | 107±3 |
| CQL | 38±2 | 39±5 | 107±9 | 10±2 | 0.4±0.2 | 101±2 | 4±1 | 2±1 | 87±5 | 0.2±0.1 | -0.1±0.1 | 95±5 |
| IDQL | 76±10 | 64±7 | 140±6 | 6±2 | 0±0 | 105±1 | 2±1 | 2±1 | 125±4 | 0±0 | -0±0 | 107±1 |
| SRPO | 69±7 | 61±7 | 134±4 | 3±3 | 0±0 | 105±0.5 | 1±1 | 2±1 | 127±0 | 0±0 | -0±0 | 106±2 |
| CAC | 64±8 | 56±10 | 103±9 | 5±2 | 1±0 | 98±3 | 2±0 | 1±1 | 92±11 | 0±0 | -0±0 | 93±6 |
| FAWAC | 67±5 | 62±10 | 118±6 | 2±1 | 0±1 | 103±1 | 2±1 | 1±0 | 118±3 | 0±0 | -0±0 | 105±3 |
| FBRAC | 77±7 | 67±9 | 119±7 | 4±2 | 0±0 | 104±2 | 2±1 | 2±1 | 119±9 | 0±0 | 1±1 | 105±2 |
| IFQL | 71±12 | 80±11 | 139±5 | 7±2 | 2±2 | 104±2 | 3±1 | 2±1 | 117±9 | 0±0 | -0±0 | 104±3 |
| FQL | 53±16 | 74±11 | 142±6 | 0±0 | 2±1 | 104±1 | 1±1 | 11±9 | 125±3 | 0±0 | -0±0 | 107±1 |
| PQL(Ours) | 82±3 | 87±6 | 148±7 | 4±1 | 3±0.8 | 107±2 | 4±1 | 14±6 | 134±1 | 0.8±0.1 | 0.5±0.1 | 110±1 |

*Table 3.* Algorithm comparison on selected OGBench tasks, grouped by environment. Red indicates the best results while blue indicates the second best.

| | Antmaze | | Humanoid | | Antsoccer | Cube | | Scene | Puzzle | |
|---|---|---|---|---|---|---|---|---|---|---|
| Algorithm | Large-nav | Giant-nav | Medium-nav | Large-nav | Arena-nav | Single-play | Double-play | Play | 3x3-Play | 4x4-Play |
| BC | 0±0 | 0±0 | 1±0 | 0±0 | 1±0 | 2±1 | 0±0 | 1±1 | 1±1 | 0±0 |
| IQL | 48±20 | 0±0 | 32±7 | 0±0 | 3±2 | 80±8 | 1±0 | 12±8 | 2±1 | 5±2 |
| CQL | 23±12 | 0±0 | 20±4 | 0±0 | 1±1 | 82±7 | 2±1 | 8±2 | 1±1 | 4±2 |
| IDQL | 0±0 | 0±0 | 1±2 | 0±0 | 5±2 | 96±1 | 16±10 | 30±14 | 3±3 | 26±6 |
| SRPO | 0±0 | 0±0 | 0±0 | 0±0 | 20±7 | 96±2 | 0±0 | 2±2 | 0±0 | 7±8 |
| CAC | 42±57 | 0±0 | 38±19 | 1±0 | 0±0 | 88±12 | 2±2 | 32±18 | 1±0 | 1±0 |
| FAWAC | 1±1 | 0±0 | 6±2 | 0±0 | 12±3 | 81±10 | 2±1 | 18±6 | 1±2 | 0±0 |
| FBRAC | 70±20 | 0±0 | 25±6 | 0±0 | 24±4 | 83±8 | 22±12 | 46±20 | 2±2 | 5±3 |
| IFQL | 24±17 | 1±0 | 69±11 | 6±12 | 16±9 | 79±5 | 9±15 | 0±0 | 0±0 | 21±11 |
| FQL | 80±32 | 4±4 | 19±13 | 7±2 | 39±36 | 98±3 | 36±6 | 76±9 | 16±5 | 11±3 |
| PQL(Ours) | 84±7 | 7±1 | 30±10 | 9±1 | 41±15 | 100±1 | 41±4 | 80±3 | 19±3 | 16±1 |

a visible performance gap over the divergence-only variant throughout training.

On these harder tasks, "PQL with Div" tends to plateau at a lower mean score, suggesting that controlling only the expansion and compression of the flow can be insufficient to prevent convergence to a suboptimal solution. The cross-seed variance of the two variants is similar on these benchmarks, so the qualitative differences lie primarily in mean performance rather than in run-to-run stability. Overall, the experiment supports our design choice: constraining the complete local geometry of the flow, including shear and rotation in addition to expansion and compression, provides a more dependable regularizer for offline RL than divergence alone. We provide further evidence on Adroit and OGBench in Section D.

## 5.4. Ablation Study

To dissect the individual contributions of our two primary components (the PDE-guided regularizer and the adaptive Beta sampling), we conduct a targeted ablation study. We evaluate three distinct variants: PQL (Ours), the full pro-

posed algorithm with both components; PQL-Beta, a variant using the PDE regularizer but with standard uniform timestep sampling; and PQL-PDE, a variant using our adaptive Beta sampling but without the PDE regularizer.

Our ablation study, presented in Table 4[1], provides critical insights into the interplay between our proposed components. The performance of the PQL-Beta variant, which applies the PDE regularizer with standard uniform time sampling, is particularly revealing. This model struggles across several environments, suggesting that adding the PDE constraint without a more efficient learning mechanism can be detrimental. While the regularizer successfully creates a smoother optimization landscape, the unfocused updates from uniform sampling appear insufficient for the policy to find a high-performing solution within this more challenging, constrained space. In contrast, the PQL-PDE variant, which uses adaptive Beta sampling without the regularizer, achieves strong results. This indicates that focusing the training on the most informative parts of the generative trajectory is a powerful mechanism for improving perfor-

---

[1]For more details and results please refer to Section D.2

*Table 4.* Ablation study of PQL components on D4RL Gym-MuJoCo environments. We compare our full method against variant without the PDE regularizer (PQL-PDE) and without Beta sampling (PQL-Beta).

| | HalfCheetah | | | Hopper | | | Walker2d | | |
|---|---|---|---|---|---|---|---|---|---|
| **Algorithm** | **Medium-Expert** | **Medium** | **Medium-Replay** | **Medium-Expert** | **Medium** | **Medium-Replay** | **Medium-Expert** | **Medium** | **Medium-Replay** |
| PQL-Beta (w/o Beta) | 76.2±5.5 | 39.5±2.8 | 48.1±1.5 | 98.7±4.1 | 75.3±2.5 | 85.0±4.2 | 105.6±2.1 | 79.8±4.0 | 78.5±2.9 |
| PQL-PDE (w/o PDE) | 84.3±4.9 | 45.1±3.5 | 45.2±2.4 | 102.4±3.8 | 79.5±1.9 | 88.6±3.9 | 109.8±1.9 | 82.0±3.5 | 80.1±2.1 |
| PQL | 90.2±1.4 | 54.1±2.2 | 51.3±0.3 | 111.8±2.9 | 84.2±1.1 | 92.1±3.4 | 114.4±0.8 | 86.2±3.1 | 84.4±1.2 |

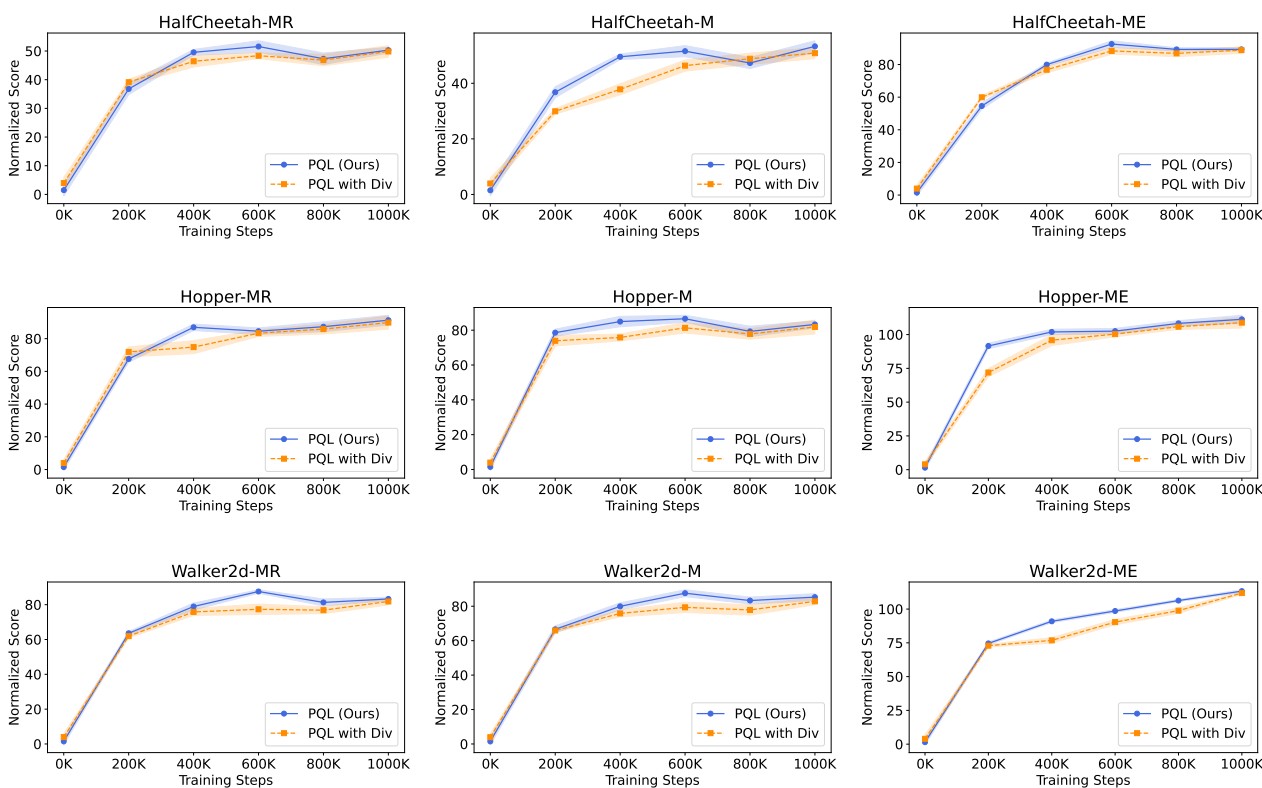

*Figure 1.* Performance comparison for different design choices of the regularizer on the D4RL Gym-MuJoCo environments. Blue line indicates the proposed PQL while the orange line indicates the variant with a divergence-only penalty.

mance, even without explicit regularization of the vector field's geometry.

### 5.5. Hyper-parameter

PQL introduces only two new hyperparameters beyond those inherited from FQL (Park et al., 2025b): the PDE coefficient $\lambda$ and the Beta shape parameter $\alpha$. The flow-matching loss weight, the distillation coefficient, and the Q-guidance coefficient are taken directly from FQL without retuning. We fix $\lambda = 0.015$ and $\alpha = 3$ across all 31 environments tested, with no per-task tuning, and the studies in this section and in Section D.4 confirm that PQL is robust to both choices within broad ranges ($\alpha \in [2.0, 5.0]$ and $\lambda \in [0.01, 0.02]$). Crucially, because the PDE regularizer

acts only on the BC-flow branch (Equation (6)) while $Q_\phi$ enters the actor objective only through the one-step prediction $a'_1$ (Equation (9)), $\lambda$ does not create a tradeoff with policy improvement; it balances flow-matching fidelity against transport smoothness within the BC-flow network alone.

To analyze the sensitivity of PQL to the Beta distribution's shape parameter $\alpha$, we conduct a hyperparameter sweep on three representative D4RL Gym-MuJoCo environments. The results, shown in Figure 2, demonstrate a clear and consistent trend across all three tasks.

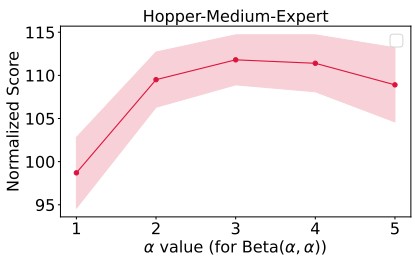 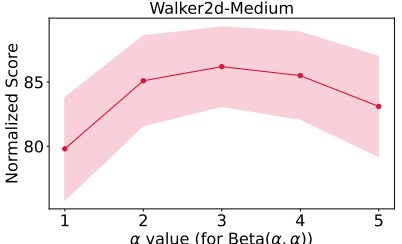 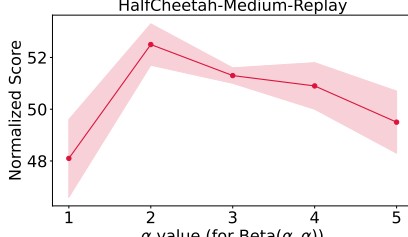

*Figure 2.* Hyper-parameter study of the Beta shape parameter $\alpha$ on three D4RL Gym-MuJoCo environments. From left to right: Hopper-Medium-Expert, Walker2d-Medium, HalfCheetah-Medium-Replay.

## 6. Conclusion

We introduced PQL, a flow-based algorithm for offline RL that addresses an RL-specific failure mode of flow-matching policies: critic-guided policy improvement deforms the transport map of the BC-flow, pushing samples toward weakly-supported regions and degrading the policy-improvement signal that downstream actors rely on. Our solution has two components. The first is a Jacobian-based regularizer motivated by the continuity equation, which acts as a tractable surrogate for transport regularity. The second is a Beta-distributed timestep sampling strategy that concentrates training on the mid-trajectory segments most affected by this regularizer. Across the D4RL, Adroit, and OGBench benchmarks, PQL achieves state-of-the-art performance, with the largest gains on the most challenging tasks where unconstrained flows are most vulnerable to Q-induced deformation.

**Limitations and future work.** Several limitations of our approach suggest natural directions for future work. First, although Hutchinson's trace estimator avoids materializing the full Jacobian, its cost still scales linearly with the action dimension through Jacobian-vector products; truly high-dimensional action spaces may benefit from more efficient stochastic estimators. Second, our Beta sampling uses a fixed shape parameter $\alpha$ that does not adapt during training. Adaptive or curriculum-based schedules that respond to changing flow complexity could in principle yield further gains. Third, the Wasserstein stability bound in Theorem 4.1 has exponential dependence on the Jacobian bound $J$ and serves to motivate Jacobian control rather than to provide a tight performance guarantee. Beyond these, extending the continuity-regularization perspective to other generative policy families, such as diffusion or score-based models, is a promising direction.

## Impact Statement

This work advances the field of generative modeling for decision-making by introducing PDE-regularized Q-Learning (PQL), an algorithm that improves the stability and reliability of autonomous policies. Through a Jacobian-based regularizer that controls the local geometry of the learned generative flow, combined with a Beta-distributed timestep sampling strategy, PQL produces more robust performance on complex control tasks such as robotics and continuous manipulation. These technical improvements facilitate the deployment of AI in safety-critical domains where unpredictable behavior is a major bottleneck. Furthermore, the method's efficiency in learning from static, offline datasets promotes more sustainable and accessible AI development by reducing the need for expensive or dangerous real-world data collection.

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

# A. Connection to the Fokker-Planck Equation

In this section, we detail the connection between the Ordinary Differential Equation (ODE) that defines our generative flow, the corresponding stochastic process, and the Fokker-Planck Equation (FPE) (Risken, 1989) that governs the evolution of its probability density. This connection provides the theoretical grounding for our PDE-guided regularizer.

## A.1. From ODEs to SDEs and Probability Flows

Our flow-matching policy learns a deterministic process governed by an ODE:

$$da_t = u_\theta(s, a_t, t)dt, \tag{11}$$

where $u_\theta$ is the learned vector field. This describes the trajectory of a single sample. To understand the evolution of the entire probability density $p_t(a|s)$, it is useful to consider the corresponding stochastic differential equation (SDE). A deterministic ODE like ours can be seen as a special case of an SDE with zero diffusion:

$$da_t = u_\theta(s, a_t, t)dt + \sigma(t)dW_t, \quad \text{where} \quad \sigma(t) = 0. \tag{12}$$

Here, $W_t$ is a standard Wiener process (Brownian motion). This SDE formulation provides a powerful bridge to the language of probability flows.

## A.2. The Fokker-Planck Equation

For any process described by an SDE of the form $dx_t = f(x_t, t)dt + g(x_t, t)dW_t$, the evolution of its probability density $p(x, t)$ is governed by a partial differential equation known as the Fokker-Planck Equation (also called the forward Kolmogorov equation):

$$\frac{\partial p(x, t)}{\partial t} = -\nabla_x \cdot [f(x, t)p(x, t)] + \frac{1}{2}\nabla_x^2 \cdot [g(x, t)^2 p(x, t)]. \tag{13}$$

The first term on the right-hand side is the drift term, describing how the mean of the distribution is pushed by the vector field. The second is the diffusion term, describing how the distribution spreads out due to noise.

In our specific case, the process is deterministic. The drift is given by our learned vector field, $f(a_t, t) = u_\theta(s, a_t, t)$, and the diffusion is zero, $g(a_t, t) = 0$. When we substitute these into the general FPE, the entire diffusion term vanishes. This leaves us with:

$$\frac{\partial p_t(a|s)}{\partial t} = -\nabla_a \cdot [u_\theta(s, a, t)p_t(a|s)]. \tag{14}$$

Rearranging this equation yields the continuity equation, which we use to motivate our regularizer in the main paper:

$$\frac{\partial p_t(a|s)}{\partial t} + \nabla_a \cdot [u_\theta(s, a, t)p_t(a|s)] = 0. \tag{15}$$

This derivation formally shows that the continuity equation is the specific instance of the Fokker-Planck Equation that governs the probability flow of a deterministic, ODE-based generative model. The continuity equation is therefore the natural analytical framework for understanding how the local geometry of $u_\theta$ shapes density evolution, which is precisely what our Jacobian regularizer is designed to constrain.

# B. Proof of Theorem 4.1

We begin by stating the integral form of Grönwall's inequality, which is a foundational result for bounding functions that satisfy certain differential inequalities.

**Lemma B.1** (Grönwall's Inequality Gronwall (1919)). *Let $y(t)$ and $\alpha(t)$ be non-negative continuous functions for $t \geq 0$, and let $\beta \geq 0$ be a constant. If $y(t)$ satisfies:*

$$y(t) \leq \beta + \int_0^t \alpha(s)y(s)\,ds$$

*Then, for all $t \geq 0$, $y(t)$ is bounded by:*

$$y(t) \leq \beta \exp\left(\int_0^t \alpha(s)\,ds\right)$$

This lemma is essential for deriving bounds on quantities that evolve over time, such as the Wasserstein distance between two probability flows.

Next, we formally establish the connection between the Frobenius norm of the Jacobian, $\|\nabla_a u\|_F$, and the two key properties of the vector field: its Lipschitz constant $L$ and the magnitude of its divergence $|\nabla_a \cdot u|$.

**Lemma B.2** (Jacobian Norm Bounds). *Let $u : \mathbb{R}^d \to \mathbb{R}^d$ be a differentiable vector field. If $\|\nabla_a u(a)\|_F \leq J$ for all $a$, then:*

1. *$u$ is $J$-**Lipschitz continuous**.*

2. *The divergence is bounded: $|\nabla_a \cdot u(a)| \leq \sqrt{d}J$.*

*Proof.* **1. Lipschitz Constant:** By the Mean Value Theorem for vector-valued functions, for any two points $a_1, a_2 \in \mathbb{R}^d$:

$$\|u(a_1) - u(a_2)\| \leq \sup_{c \in [a_1, a_2]} \|\nabla_a u(c)\|_{op} \|a_1 - a_2\|$$

where $\| \cdot \|_{op}$ is the operator norm (largest singular value). We know the operator norm is bounded by the Frobenius norm: $\|\mathbf{A}\|_{op} \leq \|\mathbf{A}\|_F$. Since $\|\nabla_a u(a)\|_F \leq J$ for all $a$, we have:

$$\|u(a_1) - u(a_2)\| \leq J\|a_1 - a_2\|$$

Thus, $u$ is $J$-Lipschitz.

**2. Divergence Bound:** The divergence is the trace of the Jacobian: $\nabla_a \cdot u = \text{Tr}(\nabla_a u)$. Let $\lambda_1, \ldots, \lambda_d$ be the eigenvalues of $\nabla_a u$. Then $\text{Tr}(\nabla_a u) = \sum_{i=1}^d \lambda_i$. By the Cauchy-Schwarz inequality:

$$|\nabla_a \cdot u|^2 = \left|\sum_{i=1}^d \lambda_i\right|^2 \leq \left(\sum_{i=1}^d |\lambda_i|^2\right)\left(\sum_{i=1}^d 1^2\right) = d\sum_{i=1}^d |\lambda_i|^2$$

Schur's inequality states that $\sum_{i=1}^d |\lambda_i|^2 \leq \|\nabla_a u\|_F^2$. Therefore:

$$|\nabla_a \cdot u|^2 \leq d\|\nabla_a u\|_F^2 \leq dJ^2$$

Taking the square root gives $|\nabla_a \cdot u| \leq \sqrt{d}J$. This completes the proof of the lemma. $\square$

*Proof.* We now prove the main theorem. Let $\rho_t^\theta$ and $\rho_t^\star$ be the probability densities induced by the vector fields $u_\theta$ and $u_\star$, respectively, starting from the same initial distribution $\rho_0$. A standard result for flows (Theorem 5.34 in Santambrogio (2015)) states that the 2-Wasserstein distance $y(t) = W_2(\rho_t^\theta, \rho_t^\star)$ satisfies the differential inequality:

$$\frac{d}{dt}y(t) \leq Ly(t) + \left(\mathbb{E}_{a \sim \rho_t^\star}[\|u_\theta(a,t) - u_\star(a,t)\|^2]\right)^{1/2}$$

where $L$ is an upper bound on the Lipschitz constants of the fields. A more refined bound, considering both expansion and contraction, is:

$$\frac{d}{dt}y(t) \leq (L + D)y(t) + E(t)$$

where $D$ bounds the divergence and $E(t) = \left(\mathbb{E}_{a \sim \rho_t^\star}[\|u_\theta(a,t) - u_\star(a,t)\|^2]\right)^{1/2}$.

From Lemma 2, we can set $L = J$ and $D = \sqrt{d}J$ (assuming $J$ is an upper bound for both fields). This gives the differential inequality:

$$\frac{d}{dt}y(t) \leq (J + \sqrt{d}J)y(t) + E(t)$$

Since $y(0) = W_2(\rho_0, \rho_0) = 0$, applying Grönwall's inequality (or solving the associated linear ODE) yields:

$$y(t) \leq \int_0^t E(\tau) \exp\left(\int_\tau^t (J + \sqrt{d}J)ds\right) d\tau$$

$$y(t) \leq \int_0^t E(\tau) \exp\left((J + \sqrt{d}J)(t - \tau)\right) d\tau$$

Since $t - \tau \leq t$ for $\tau \in [0, t]$, we can find a looser but simpler bound by taking the maximal value of the exponential term outside the integral:

$$W_2(\rho_t^\theta, \rho_t^\star) \leq \exp((J + \sqrt{d}J)t) \int_0^t \left(\mathbb{E}_{a \sim \rho_\tau^\star}[\|u_\theta(s, a, \tau) - u_\star(s, a, \tau)\|^2]\right)^{1/2} d\tau$$

This completes the proof of the theorem. $\qquad\square$

## C. Proof of Theorem 4.2

*Proof.* The proof follows the standard procedure for finding an optimal importance sampling distribution. The goal is to minimize the variance of an unbiased estimator subject to the constraint that the sampling distribution is a valid probability density function.

We wish to estimate the total gradient of the loss, which is an integral over the time variable $t$:

$$G(\theta) = \nabla_\theta \mathcal{L}(\theta) = \nabla_\theta \int_0^1 \ell(t)dt = \int_0^1 \nabla_\theta \ell(t)dt$$

We can form a stochastic, single-sample Monte Carlo estimator for this integral by sampling a timestep $t$ from a proposal distribution $\pi(t)$ and weighting the result. The importance sampling estimator for the gradient $G(\theta)$ is:

$$\hat{G}(t) = \frac{\nabla_\theta \ell(t)}{\pi(t)}$$

This is an **unbiased estimator**, as its expectation is equal to the true gradient:

$$\mathbb{E}_{t \sim \pi(t)}[\hat{G}(t)] = \int_0^1 \frac{\nabla_\theta \ell(t)}{\pi(t)} \pi(t)dt = \int_0^1 \nabla_\theta \ell(t)dt = G(\theta)$$

Our goal is to choose the distribution $\pi(t)$ that minimizes the variance of this estimator. The variance of a vector estimator is the expected squared Euclidean distance from its mean. Minimizing this is equivalent to minimizing $\mathbb{E}[\|\hat{G}(t)\|^2]$, since the mean $G(\theta)$ is fixed with respect to $\pi(t)$. The second moment is:

$$\mathbb{E}_{t \sim \pi(t)}[\|\hat{G}(t)\|^2] = \int_0^1 \left\|\frac{\nabla_\theta \ell(t)}{\pi(t)}\right\|^2 \pi(t)dt = \int_0^1 \frac{\|\nabla_\theta \ell(t)\|^2}{\pi(t)} dt$$

So, our optimization problem is to find the function $\pi(t)$ that minimizes this integral.

We must minimize the variance expression subject to the constraint that $\pi(t)$ is a valid probability distribution, meaning $\int_0^1 \pi(t)dt = 1$. This is a constrained optimization problem that we can solve using a **Lagrange multiplier** $\lambda$. The Lagrangian $\mathcal{J}$ is:

$$\mathcal{J}(\pi, \lambda) = \int_0^1 \frac{\|\nabla_\theta \ell(t)\|^2}{\pi(t)} dt + \lambda \left(\int_0^1 \pi(t)dt - 1\right)$$

To find the optimal $\pi(t)$, we take the functional derivative of $\mathcal{J}$ with respect to $\pi(t)$ and set it to zero:

$$\frac{\delta \mathcal{J}}{\delta \pi(t)} = -\frac{\|\nabla_\theta \ell(t)\|^2}{\pi(t)^2} + \lambda = 0$$

Solving for $\pi(t)$, we find:

$$\pi(t)^2 = \frac{\|\nabla_\theta \ell(t)\|^2}{\lambda} \implies \pi(t) = \frac{\|\nabla_\theta \ell(t)\|}{\sqrt{\lambda}}$$

This shows that the optimal distribution $\pi(t)$ must be proportional to the norm of the instantaneous gradient $\|\nabla_\theta \ell(t)\|$.

We find the value of the constant $\sqrt{\lambda}$ by enforcing the constraint $\int_0^1 \pi(t)dt = 1$:

$$\int_0^1 \frac{\|\nabla_\theta \ell(t)\|}{\sqrt{\lambda}} dt = 1$$

$$\frac{1}{\sqrt{\lambda}} \int_0^1 \|\nabla_\theta \ell(t)\| dt = 1 \implies \sqrt{\lambda} = \int_0^1 \|\nabla_\theta \ell(\tau)\| d\tau$$

Let $Z = \int_0^1 \|\nabla_\theta \ell(\tau)\| d\tau$ be the normalization constant. Substituting this back, we get the full form of the optimal distribution:

$$\pi^*(t) = \frac{\|\nabla_\theta \ell(t)\|}{Z} = \frac{\|\nabla_\theta \ell(t)\|}{\int_0^1 \|\nabla_\theta \ell(\tau)\| d\tau}$$

This shows that the sampling distribution $\pi^*(t)$ that minimizes the variance of the stochastic gradient estimate is the one that is proportional to the magnitude of the gradient being estimated at each point $t$. □

## D. Extra Experiments

### D.1. Design Choice

We also provide the extra experiments regarding the design choice in Adroit and OGBench. Figure 3 presents the extra ablation study comparing our proposed method, PQL (Ours), against a key variant, PQL with Divergence. This variant replaces our Jacobian-based regularizer with a divergence-only penalty, allowing us to isolate the benefits of constraining the full local geometry of the velocity field (including shear and rotation) rather than only its divergence. The comparison is conducted across two distinct and challenging benchmark suites: Adroit and OGBench.

Across the Adroit manipulation tasks (Figure 3a), PQL shows a consistent advantage. In most of the 12 environments, our method achieves a higher mean normalized score than the divergence-based variant. The performance gap is particularly notable in complex, expert-level datasets such as `Pen-Expert` and `Hammer-Expert`, suggesting that constraining the full Jacobian helps when modeling intricate, high-skill behaviors.

This trend of robust outperformance continues on the more diverse OGBench suite (Figure 3b), which includes challenging locomotion and navigation tasks. In environments like `Humanoid-Medium`, `Cube-Double`, and `Puzzle-4x4`, PQL again establishes a significant performance advantage. The consistent success across 22 distinct environments strongly indicates that our proposed regularization is a more effective and generalizable approach than the standard divergence-based alternative.

### D.2. Ablation Study

*Table 5.* Ablation Study in Adroit environment. We ablate our method by removing the PDE regularizer (PQL-PDE) and the Beta-distributed time sampling (PQL-Beta)

| | Pen | | | Door | | | Hammer | | | Relocate | | |
|---|---|---|---|---|---|---|---|---|---|---|---|---|
| Algorithm | Human | Cloned | Expert | Human | Cloned | Expert | Human | Cloned | Expert | Human | Cloned | Expert |
| PQL-Beta (w/o Beta) | 75±8 | 80±10 | 140±9 | 2±2 | 1±1 | 100±5 | 2±2 | 10±8 | 128±6 | 0.5±0.2 | 0.2±0.2 | 105±3 |
| PQL-PDE (w/o PDE) | 53±12 | 72±11 | 143±4 | 0±0 | 2±1 | 102±2 | 1±1 | 11±9 | 123±3 | 0±0 | 0±0 | 106±1 |
| PQL | 82±3 | 87±6 | 148±7 | 4±1 | 3±0.8 | 107±2 | 4±1 | 14±6 | 134±1 | 0.8±0.1 | 0.5±0.1 | 110±1 |

*Table 6.* Ablation Study on selected OGBench tasks, grouped by environment.

| | Antmaze | | Humanoid | | Antsoccer | Cube | | Scene | Puzzle | |
|---|---|---|---|---|---|---|---|---|---|---|
| Algorithm | Large-nav | Giant-nav | Medium-nav | Large-nav | Arena-nav | Single-play | Double-play | Play | 3x3-Play | 4x4-Play |
| PQL-Beta (w/o Beta) | 75±10 | 4±2 | 18±9 | 7±0.6 | 37±13 | 83±1 | 34±2 | 70±2 | 15±3 | 13±1 |
| PQL-PDE (w/o PDE) | 77±28 | 5±3 | 20±11 | 8±1 | 40±31 | 97±3 | 35±7 | 77±8 | 17±4 | 12±2 |
| PQL | 84±7 | 7±1 | 30±10 | 9±1 | 41±15 | 100±1 | 41±4 | 80±3 | 19±3 | 16±1 |

To further validate the individual contributions of our proposed components, we conduct a thorough ablation study, with results presented in Table 5 for the Adroit suite and Table 6 for the OGBench suite. We systematically evaluate the impact of our two core contributions: the continuity-based PDE regularizer and the Beta-distributed time sampling strategy.

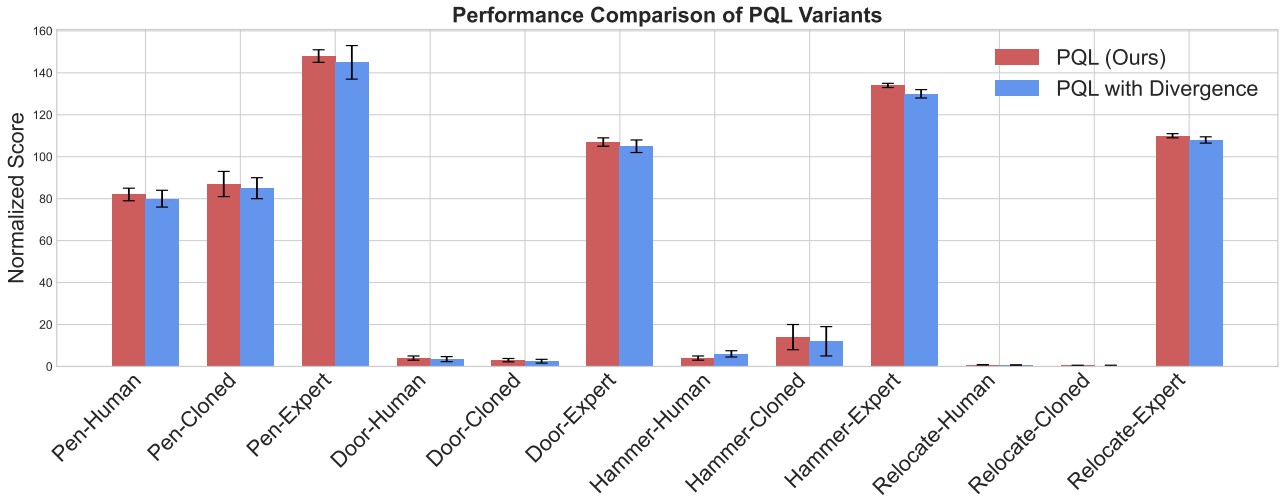

*(a)* Performance comparison across Adroit tasks.

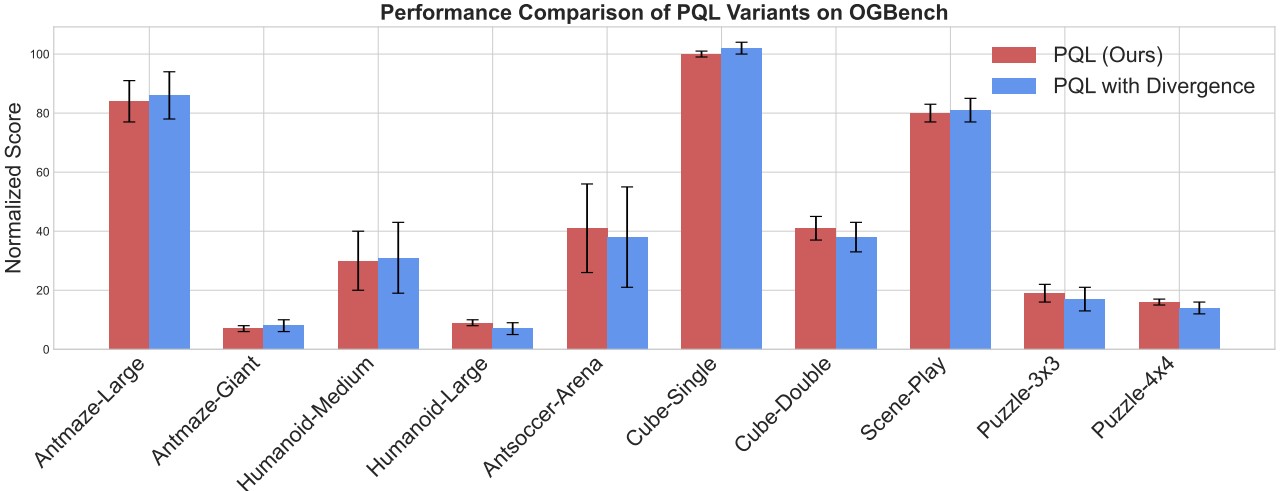

*(b)* Performance comparison across OGBench tasks.

*Figure 3.* Performance comparison of PQL (Ours) and PQL with Divergence across two benchmark suites: (a) Adroit and (b) OGBench. Error bars represent one standard deviation over multiple seeds.

**Impact of the PDE Regularizer.** We first analyze the impact of our central contribution by comparing PQL against the **PQL-PDE (w/o PDE)** variant, which removes the regularizer. The results unequivocally demonstrate that the PDE constraint is critical for performance. Across the Adroit tasks (Table 5), removing the regularizer causes a substantial performance collapse, especially on the challenging human and cloned datasets such as `Pen-Human` (82 vs. 53) and `Hammer-Cloned` (14 vs. 11). This trend is mirrored in the OGBench results (Table 6), where the performance gap remains significant in complex tasks like `Antmaze-Large-nav` (84 vs. 77) and `Humanoid-Medium` (30 vs. 20). This provides strong empirical evidence for our main hypothesis: that leaving the critic-guided transport map unconstrained is a key performance bottleneck in flow-based offline RL, and that controlling its local geometry via our Jacobian regularizer directly and effectively addresses this limitation.

**Impact of Beta-distributed Time Sampling.** Next, we investigate the importance of our proposed time sampling strategy by evaluating the **PQL-Beta (w/o Beta)** variant, which reverts to standard uniform sampling. The results in both tables show that this change also leads to a consistent degradation in performance. For instance, in `Pen-Cloned` (87 vs. 80) and

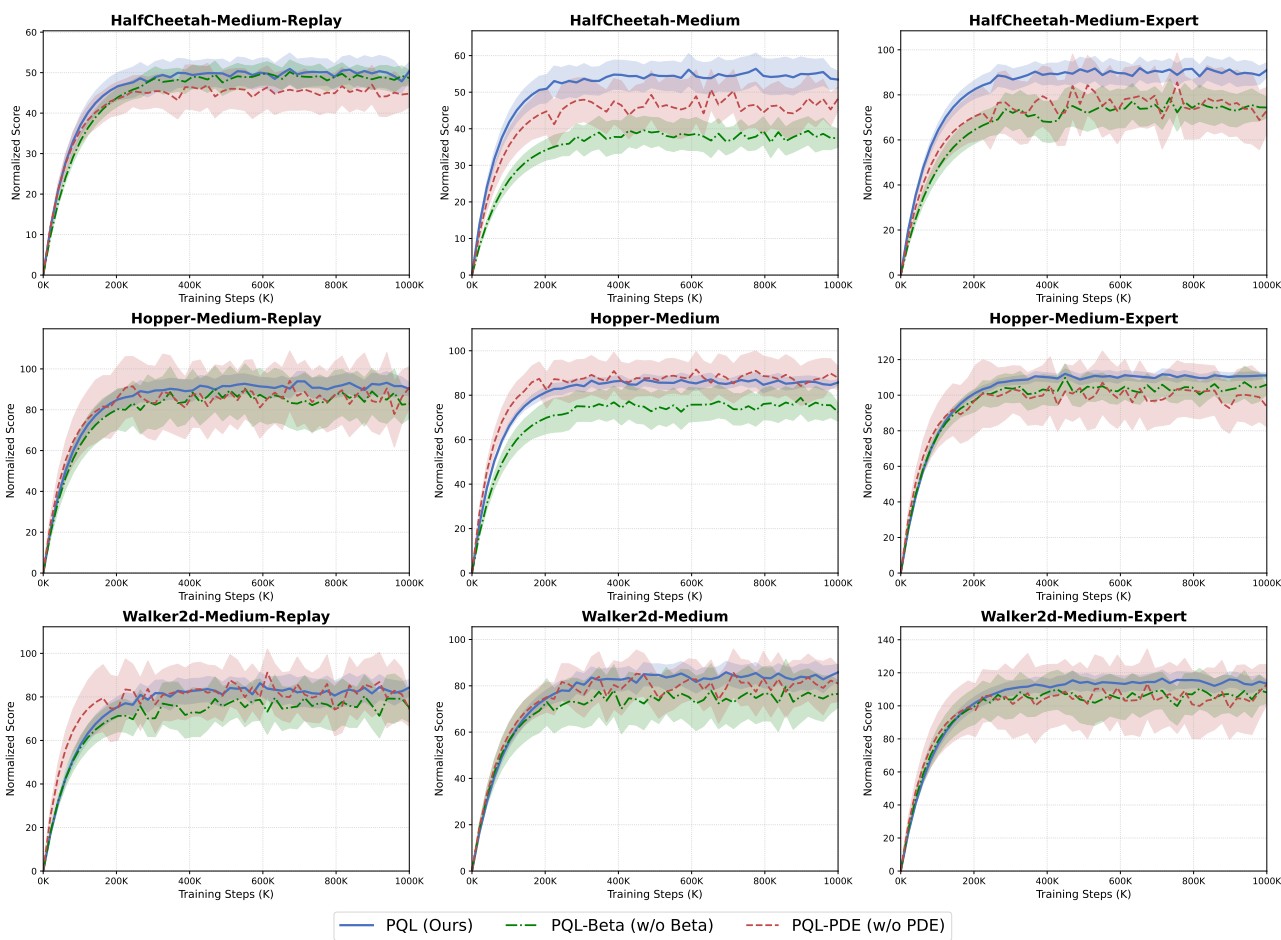

*Figure 4.* Training Curve on D4RL Gym-MuJoCo environments across 8 different seeds. Blue line indicates the proposed PQL while the red indicates the variant without PDE regularizer, the green line indicates the variant without Beta.

`Cube-Single` (100 vs. 83), the absence of Beta sampling yields significantly lower scores. This validates our second claim: that naively optimizing the PDE-constrained objective is challenging. The superior performance of our full PQL model demonstrates that Beta-distributed sampling, by concentrating updates in the most critical mid-range of the generative trajectory where data and noise are maximally entangled, provides a more stable and effective training signal. This tailored optimization strategy is crucial for unlocking the full potential of the PDE constraint. In summary, our ablation studies confirm that both the PDE regularizer and the Beta-distributed time sampling are integral to PQL's success. The removal of either component leads to a significant performance drop, affirming that our method's strong results stem from the synergy between a principled geometric constraint and an effective, tailored optimization strategy.

**Discussion** It is instructive to contextualize our contributions with respect to recent state-of-the-art flow-matching methods such as FQL. While highly effective, our experiments suggest that unconstrained flow-matching policies can exhibit significant performance variance on complex tasks. For instance, our `PQL-PDE (w/o PDE)` baseline, which is analogous to FQL, demonstrates high variance in challenging environments like `Antmaze-Large-nav` and `Antsoccer-Arena-nav`, as shown in Table 6 and Table 3. In contrast, our full PQL model not only achieves a higher mean score but also substantially reduces this variance, highlighting the stabilizing effect of our PDE regularizer on the critic-guided transport map. Furthermore, this analysis clarifies the nuanced role of our Beta-distributed time sampling strategy. Prior work (Park et al., 2025b) suggests that such sampling schemes offer little general benefit for standard flow-matching objectives, a finding that aligns with the moderate performance drop of our `PQL-Beta (w/o Beta)` ablation. However, in our setting the picture is different: the Beta distribution is not proposed as a universal performance

enhancer, but as a specific solution to the more challenging optimization landscape introduced by our PDE regularizer. The regularizer creates a more structured but harder optimization problem by penalizing high-Jacobian transport, and the Beta sampling strategy is essential to navigate it by focusing training on the mid-trajectory segments most affected by the regularizer. Therefore, the necessity of our approach lies in the synergy of its components: the PDE regularizer controls the geometry of the critic-guided transport map, and the Beta sampling makes this geometric constraint amenable to effective optimization.

### D.3. Additional Diagnostic Experiments

We collect here several diagnostic experiments and analyses that complement the main results: a cross-architecture validation on FBRAC; a 2D toy experiment that visualizes how Q-guidance deforms the BC-flow; and measurements of loss components and gradient interactions during training on real offline RL environments.

**Cross-Architecture Generalization on FBRAC.** To verify that our two contributions generalize beyond FQL, we apply the same Jacobian regularizer and Beta sampling to FBRAC (Park et al., 2025b), a structurally distinct flow-based offline RL method that uses advantage-weighted learning rather than critic-guided actor distillation. Table 7 reports the results. The pattern matches FQL: adding the PDE regularizer alone yields a small mean drop but already reduces cross-seed variance, while adding Beta sampling on top recovers and surpasses the FBRAC baseline by roughly 5 normalized points on Pen-Human and Antmaze-Large. The variance reduction is substantial: $43$–$50\%$ lower standard deviation on both environments. This confirms that the combination of PDE regularization plus Beta sampling is not specific to FQL, and that both components are needed for the gain.

*Table 7.* Cross-architecture validation on FBRAC. PDE coefficient $\lambda = 0.015$, Beta shape $\alpha = 3$. Mean $\pm$ one standard deviation over 8 seeds.

| Config | Pen–Human | Antmaze-Large |
|---|---|---|
| FBRAC (original) | $77 \pm 7$ | $70 \pm 20$ |
| FBRAC + PDE only | $74 \pm 5$ | $66 \pm 13$ |
| **FBRAC + PDE + Beta** | $\mathbf{82 \pm 4}$ | $\mathbf{75 \pm 10}$ |

**Diagnostic 2D Toy: Visualizing Critic-Induced Flow Deformation.** To make the failure mode introduced by critic guidance concrete, we set up a 2D toy offline-RL task in which an unregularized flow policy and PQL are trained with the same dataset and critic. Figure 5 visualizes the Frobenius norm of the velocity-field Jacobian $\|\nabla_a u_\theta(s, a, t)\|_F$ as a heatmap at mid-trajectory ($t = 0.5$); the unregularized flow develops a sharp Jacobian peak of $2.82$, whereas PQL keeps it small and spatially smooth at $1.64$ (a $41.8\%$ reduction). The right panel additionally overlays the $\text{Beta}(3, 3)$ density on the time axis, showing that the sampler concentrates training on precisely the timesteps where the unregularized flow is most irregular. Figure 6 summarizes the Jacobian norm across the trajectory and decomposes the training loss components for the two methods. Figure 7 visualizes the resulting ODE trajectories under Q-guidance: the unregularized flow transports nearby latent points toward weakly-supported, high-Q regions of action space, while PQL preserves a regular transport map.

**Training Dynamics on Pen-Human-v1.** We track training dynamics on the challenging `pen-human-v1` environment over 500K gradient steps. Figure 8 shows the four loss components of PQL ($\mathcal{L}_{\text{FM}}, \mathcal{L}_Q, \mathcal{L}_{\text{distill}}, \mathcal{L}_{\text{PDE}}$) over training. Two observations stand out. First, the Q-value magnitude (proxy for the critic-driven policy improvement signal) grows roughly $8\times$ as training progresses, indicating that the actor is being driven increasingly hard by the critic. Second, despite this growing Q-pressure, the Jacobian-norm regularizer $\mathcal{L}_{\text{PDE}}$ grows only $1.07\times$, demonstrating that the regularizer holds the line on transport regularity even as critic gradients intensify. This is the empirical signature on real data of the failure mode visualized in the toy task: without our regularizer, Q-pressure manifests as growing Jacobians.

Figure 9 reports the cosine similarity between $\nabla \mathcal{L}_{\text{FM}}$ and $\nabla \mathcal{L}_{\text{PDE}}$ throughout training. The cosine is mixed early but becomes predominantly negative later, consistent with the PDE term acting as a countervailing regularizer that discourages updates associated with increasingly irregular transport, rather than duplicating the FM objective. We additionally verified that $\cos(\nabla \mathcal{L}_{\text{PDE}}, \nabla \mathcal{L}_Q) = 0$ throughout training, which follows from architectural separation: the PDE regularizer acts on the BC-flow network while $Q_\phi$ enters the actor objective only through the one-step prediction $a'_1$. Consequently, the PDE coefficient $\lambda$ does not create a tradeoff with policy improvement, as discussed in Section D.4.

**Vector Field and Jacobian Norm Comparison (2D Toy with Q-Guidance)**

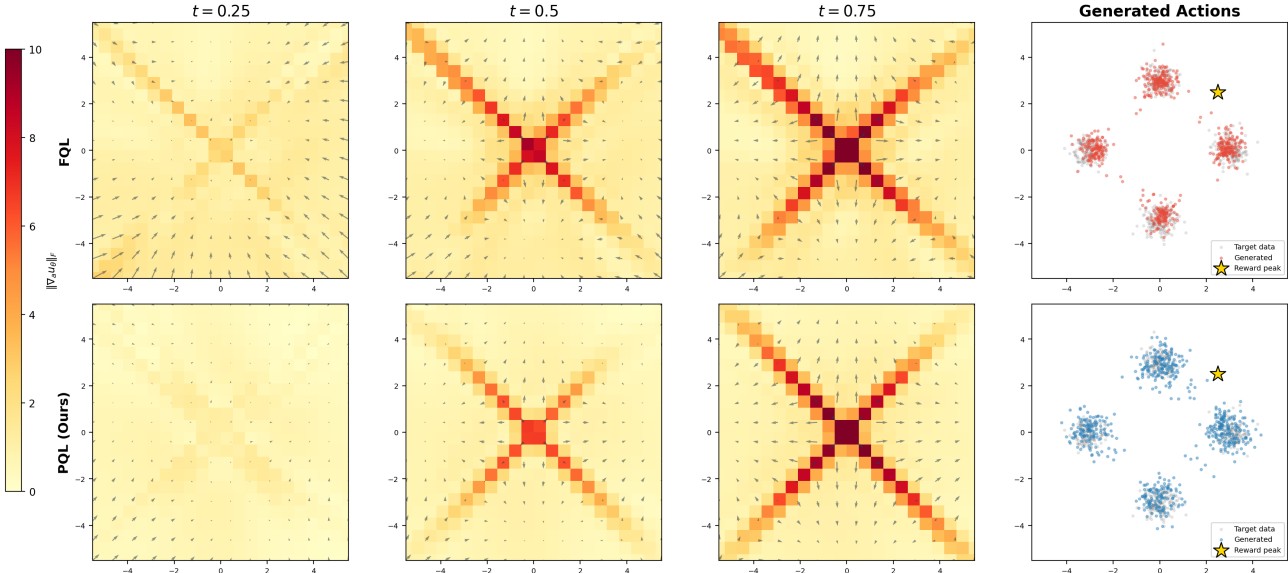

*Figure 5.* 2D toy task. Heatmaps of $\|\nabla_a u_\theta\|_F$ at $t = 0.5$ for an unregularized flow (left, peak 2.82) and PQL (right, peak 1.64). The right panel overlays the $\mathrm{Beta}(3, 3)$ density on the time axis.

**Extended Training Curves on Challenging Tasks.** Figure 10 shows training curves of PQL versus its two ablations (PQL-PDE without the regularizer, PQL-Beta without Beta sampling) on three high-variance tasks: Pen-Human, Antmaze-Large, and Antsoccer-Arena. Across all three, PQL's confidence bands are visibly tighter than those of the ablations, with the clearest variance reduction on Pen-Human and Antmaze-Large. The PQL-PDE ablation (no regularizer) shows the widest cross-seed bands on Antmaze-Large in particular, consistent with our claim that the regularizer is the primary driver of stability under critic guidance.

### D.4. Hyper-Parameters

**Beta Sampling $\alpha$** To investigate the impact of our proposed Beta-distributed time sampling strategy, we conduct a comprehensive sensitivity analysis on the key hyperparameter, $\alpha$, which controls the shape of the sampling distribution. The results across all three benchmark suites—Adroit, OGBench, and Gym-MuJoCo—are summarized in the heatmaps in Figure 11. The figure visualizes the normalized score for $\alpha$ values ranging from 1.0 (equivalent to uniform sampling) to 5.0.

The results present a clear and consistent trend across all 31 distinct environments. Optimal performance is almost universally achieved when $\alpha$ is set to either 2.0 or 3.0, as indicated by the bolded scores and highlighted cells in both heatmaps. This finding strongly confirms our central hypothesis: concentrating the sampling of the time variable $t$ in the critical mid-range of the generative trajectory, where data and noise are most entangled, provides a more effective and generalizable learning signal for the PDE-constrained objective. The consistency of this result across tasks with vastly different dynamics, from complex manipulation (Adroit) to navigation (OGBench) and locomotion (Gym-MuJoCo), underscores the robustness of our approach. Furthermore, the analysis reveals a significant performance degradation at the extremes. The notably lower scores at $\alpha = 1.0$ across nearly every task provide strong, widespread evidence against uniform time sampling, validating a core claim of our work. Similarly, a very high $\alpha$ (e.g., 5.0), which creates an overly narrow sampling peak, also tends to degrade performance. Based on this comprehensive analysis, we selected an $\alpha$ value of 3.0 for all main experiments, as it provided the most consistent and high-performing results across the full suite of environments.

**Why Beta and not another mid-concentrated distribution?** Theorem 4.2 motivates non-uniform timestep emphasis but does not uniquely specify the Beta family. To verify that Beta is a competitive choice and not merely convenient, we compared several mid-concentrated alternatives that all place additional mass near $t = 0.5$. Each variant was trained with the full PQL objective (PDE coefficient $\lambda = 0.015$) on Pen-Human and Antmaze-Large; results are reported in Table 8.

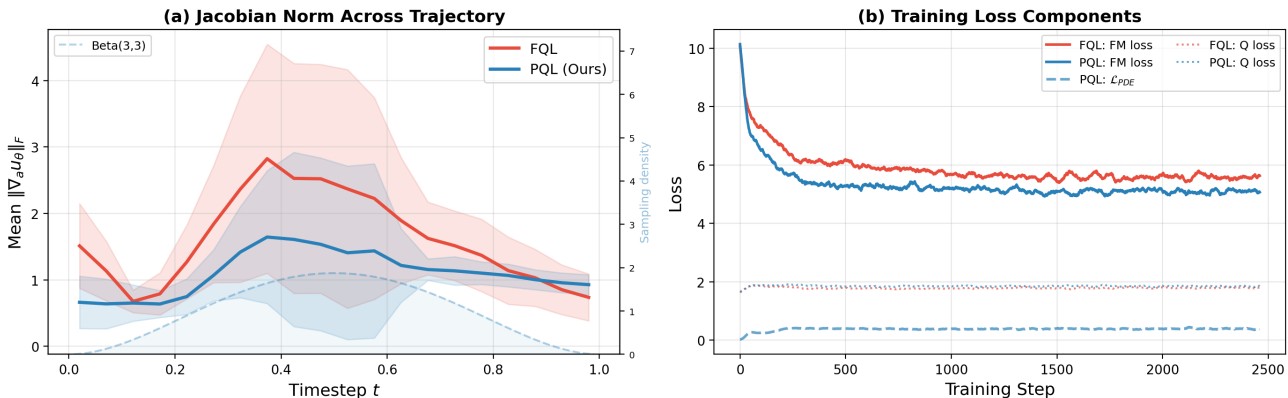

*Figure 6.* 2D toy task. Jacobian Frobenius norm of the learned velocity field across the trajectory $t \in [0, 1]$ and decomposition of the training loss components for PQL and the unregularized baseline.

*Table 8.* Comparison of timestep sampling distributions under the PQL objective ($\lambda = 0.015$) on two representative tasks, 8 seeds. Mean $\pm$ one standard deviation. Beta$(3, 3)$ outperforms uniform, truncated-normal, and logit-normal alternatives.

| Distribution + PDE | Pen-Human | Antmaze-Large |
|---|---|---|
| Uniform | $75\pm8$ | $73\pm6$ |
| Truncated normal | $74\pm5$ | $77\pm3$ |
| Logit-normal | $71\pm4$ | $72\pm6$ |
| Beta$(3, 3)$ (ours) | $\mathbf{82\pm3}$ | $\mathbf{84\pm7}$ |

Beta$(3, 3)$ outperforms all tested alternatives by a clear margin on both environments. Importantly, not all mid-concentrated distributions help: logit-normal is the worst of the four on both tasks despite also concentrating mass near $t = 0.5$, and it underperforms even uniform sampling. We attribute this to gradient interference. In our training-dynamics measurements, the gradient cosine $\cos(\nabla \mathcal{L}_{\mathrm{FM}}, \nabla \mathcal{L}_{\mathrm{PDE}})$ averages roughly $-0.89$ under logit-normal sampling (near-total opposition), while under Beta sampling it remains moderate and mixed throughout training. Our claim is therefore not that Beta is theoretically unique, but that it is the simplest symmetric one-parameter family that smoothly interpolates from uniform ($\alpha = 1$) to mid-concentrated sampling ($\alpha > 1$), and that it empirically outperforms the alternatives we tested.

**PDE Temperature $\lambda$**     In addition to the time-sampling distribution, a second critical hyperparameter is the PDE temperature, $\lambda$, which serves as the weighting coefficient for our Jacobian regularizer in the final loss function. This parameter governs the fundamental trade-off between two competing objectives: the fidelity of the terminal action distribution and the regularity of the underlying transport map. An overly large $\lambda$ could force the model to prioritize a smooth flow at the expense of accurately capturing the complexity of the target action distribution, resulting in a suboptimal policy. Intuitively, we hypothesize that a small, non-zero value for $\lambda$ is required to balance these objectives effectively. To empirically determine this balance, we perform a detailed sensitivity analysis on $\lambda$. Figure 12 presents the results of this study across all 31 environments, split into two heatmaps for clarity: (a) for Gym-MuJoCo and (b) for the combined Adroit and OGBench suites. We tested $\lambda$ values spanning several orders of magnitude to thoroughly map the performance landscape. The results provide a clear and remarkably consistent conclusion across all three distinct benchmarks. As hypothesized, the performance of PQL is highly sensitive to the value of $\lambda$, and there is a distinct optimal range. The highest scores are almost universally achieved for small values of $\lambda$, specifically within the range of $[0.01, 0.02]$, as highlighted by the cells in both figures. This demonstrates that only a small weighting is necessary for the PDE regularizer to effectively stabilize the generative path without interfering with the primary policy optimization objective. The consistency of this finding across a wide array of task complexities and dynamics speaks to the robustness of our proposed method. Furthermore, the heatmaps clearly illustrate the trade-off we aimed to balance. When $\lambda$ is too large (e.g., $\lambda = 1.0$), performance consistently and significantly degrades across every single environment. This result strongly supports our claim that overly aggressive Jacobian smoothing can harm the final policy's performance. Conversely, the strong results in the low-$\lambda$ regime confirm that the regularizer is a critical component for success. Based on this comprehensive study, we selected $\lambda = 0.015$ for all main experiments, as it represents the most consistent point of optimal performance across the full set of environments.

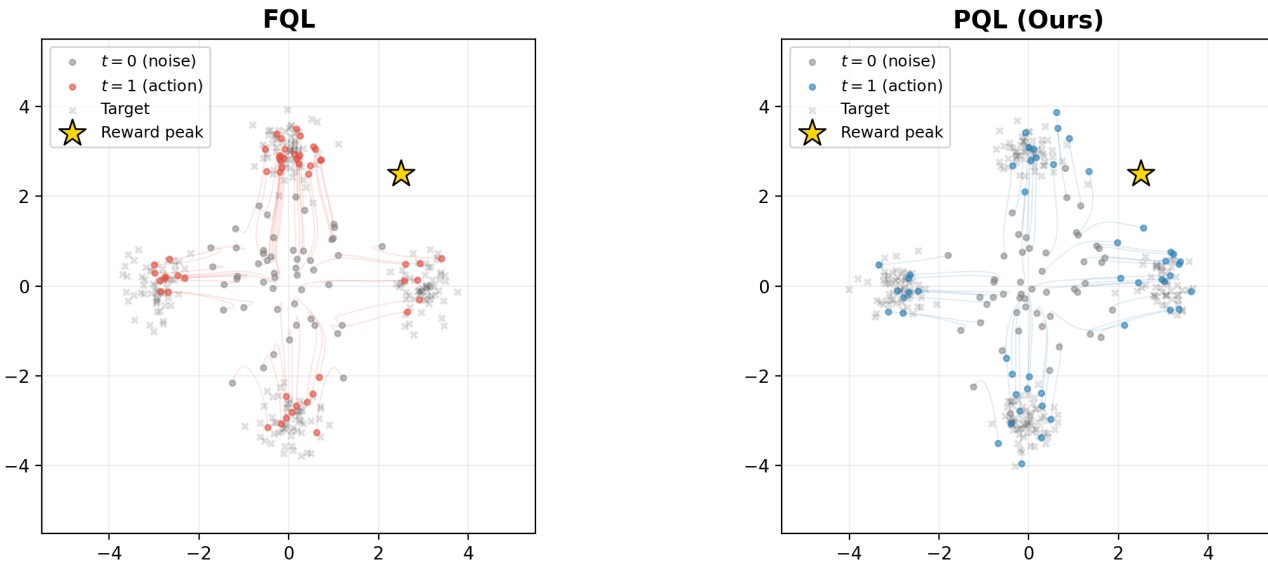

*Figure 7.* 2D toy task. ODE trajectories under Q-guidance. The unregularized flow transports nearby latent points toward weakly-supported, high-Q regions; PQL preserves a regular transport map.

## D.5. Computational Efficiency

To evaluate the computational efficiency of our method, we analyze both the theoretical complexity and the empirical wall-clock training time. A primary concern with Jacobian-based regularization is the potential cost of computing the full $D \times D$ Jacobian matrix. We circumvent this by employing Hutchinson's trace estimator (as detailed in Section 4.1), which approximates the trace using Jacobian-vector products (JVPs). This approach is highly efficient, reducing the cost to approximately one forward pass of the network per probe vector. By utilizing $K \in \{1, 2\}$ probe vectors, the total overhead of the proposed regularizer is limited to just $K$ additional forward passes per actor update. Empirically, we compare the training wall-clock time of our method (PQL) against standard baselines (FQL, Flow, CNF) and our ablation baseline (PQL-PDE) in Table 9. The comparison between PQL-PDE and the full PQL method isolates the specific cost of the $\mathcal{L}_{\mathrm{PDE}}$ regularizer. The results demonstrate that the additional overhead is minimal; for instance, PQL requires only marginally more time than the unregularized PQL-PDE and remains competitive with or faster than computationally intensive baselines like CNF. This confirms that the proposed method achieves significant gains in performance and stability with a negligible computational trade-off.

*Table 9.* Wall-clock training time (mm:ss or h:mm:ss) for PQL and selected baselines on representative tasks from Adroit and OGBench. The comparison between PQL-PDE (without the Jacobian regularizer) and the full PQL isolates the cost of $\mathcal{L}_{\mathrm{PDE}}$.

| Model | Pen-Human | Antmaze |
|---|---|---|
| FQL | 34:57 | 4:15:43 |
| Flow | 35:10 | 5:21:50 |
| CNF | 36:53 | 4:58:49 |
| PQL-PDE | 35:02 | 4:24:33 |
| PQL | 35:25 | 4:32:55 |

## E. Training and Implementation

The overall training algorithm can be found in Algorithm 1.

We also provide the complete list of hyperparameters in Table 10 for reproducibility.

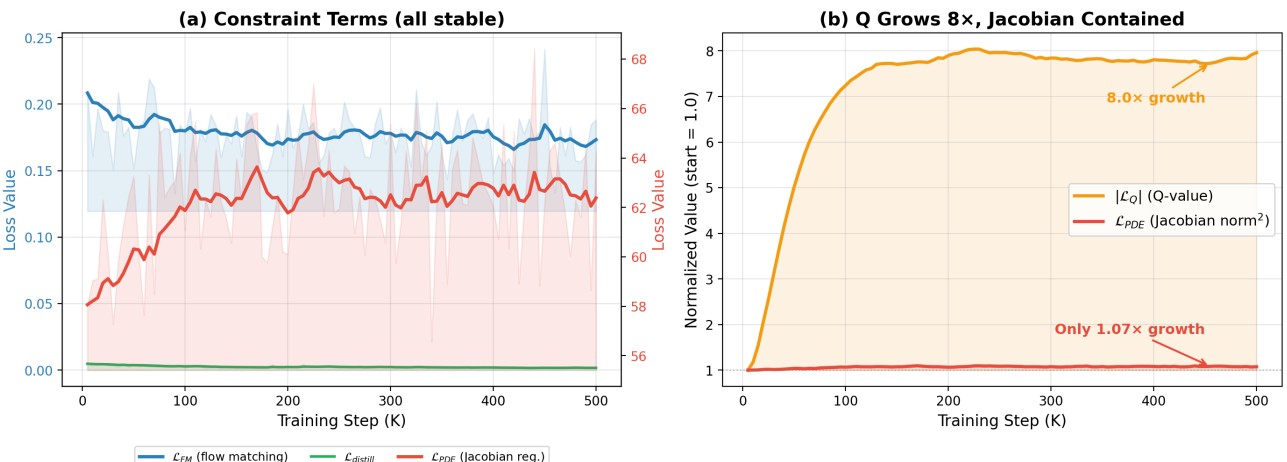

*Figure 8.* PQL training loss decomposition on `pen-human-v1` over 500K steps. Despite Q-values growing approximately $8\times$, the Jacobian penalty $\mathcal{L}_{\mathrm{PDE}}$ grows only $1.07\times$, indicating that the regularizer successfully contains critic-induced flow deformation.

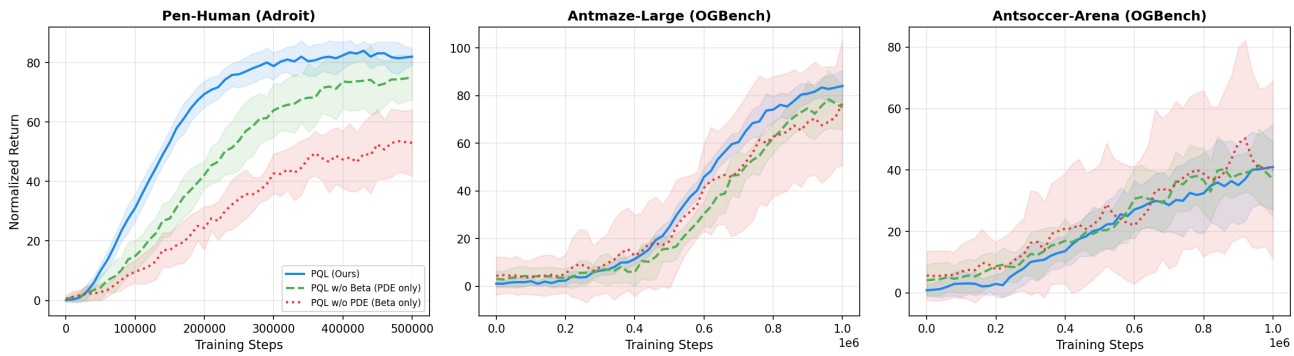

*Figure 9.* Gradient cosine similarity between $\nabla\mathcal{L}_{\mathrm{FM}}$ and $\nabla\mathcal{L}_{\mathrm{PDE}}$ over 500K training steps on `pen-human-v1`. The cosine is mixed early in training but becomes predominantly negative later, consistent with the PDE term acting as a countervailing regularizer.

*Figure 10.* Training curves with ablations (mean $\pm$ one standard deviation over 8 seeds) on Pen-Human, Antmaze-Large, and Antsoccer-Arena. PQL exhibits substantially tighter confidence bands than both ablations.

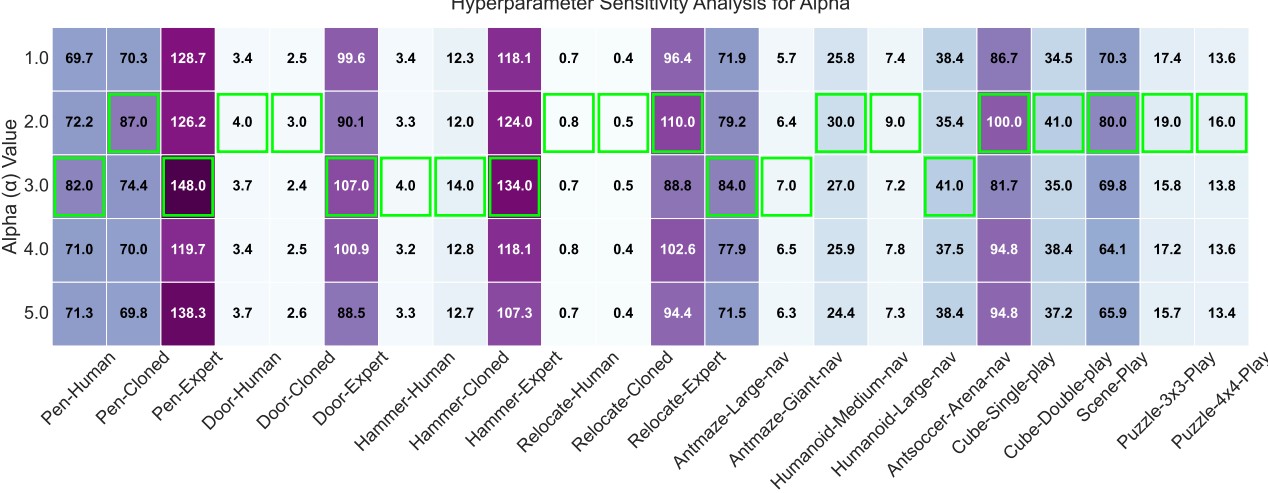

*(a)* Hyper-parameter study of $\alpha$ in the Adroit and OGBench

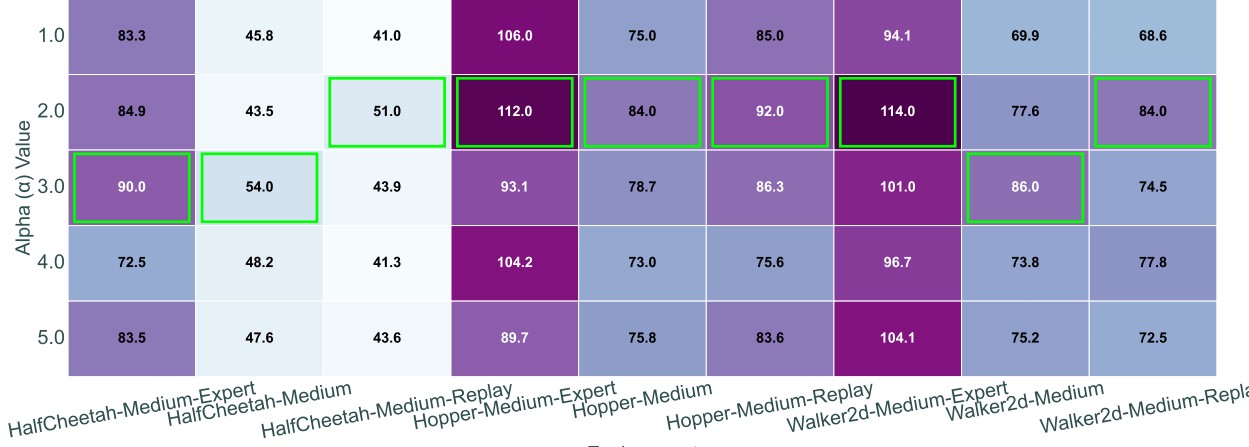

*(b)* Hyper-parameter study of $\alpha$ in the MuJoCo

*Figure 11.* Comprehensive hyper-parameter study of the Beta-distribution parameter $\alpha$ across all three benchmark suites. (a) shows results on Adroit and OGBench tasks, while (b) shows results on Gym-MuJoCo tasks.

*Table 10.* Hyperparameters for PQL.

| Hyperparameter | Value |
| --- | --- |
| Learning rate | 0.0003 |
| Optimizer | AdamW (Loshchilov & Hutter, 2017) |
| Gradient steps | 1000000 (default), 500000 (D4RL, pixel-based OGBench) |
| Minibatch size | 256 |
| MLP dimensions | [512, 512, 512, 512] |
| Nonlinearity | GELU (Hendrycks & Gimpel, 2016) |
| Target network smoothing coefficient | 0.005 |
| Discount factor $\gamma$ | 0.99 (default), 0.995 (antmaze-giant, humanoidmaze, antsoccer) |
| Flow time sampling distribution | Beta($\alpha, \alpha$) with weighted sampling |
| Beta Parameter | $\alpha = 2$ or $\alpha = 3$, please refer to Section D.4. |
| PDE coefficient $\lambda$ | 0.015 |

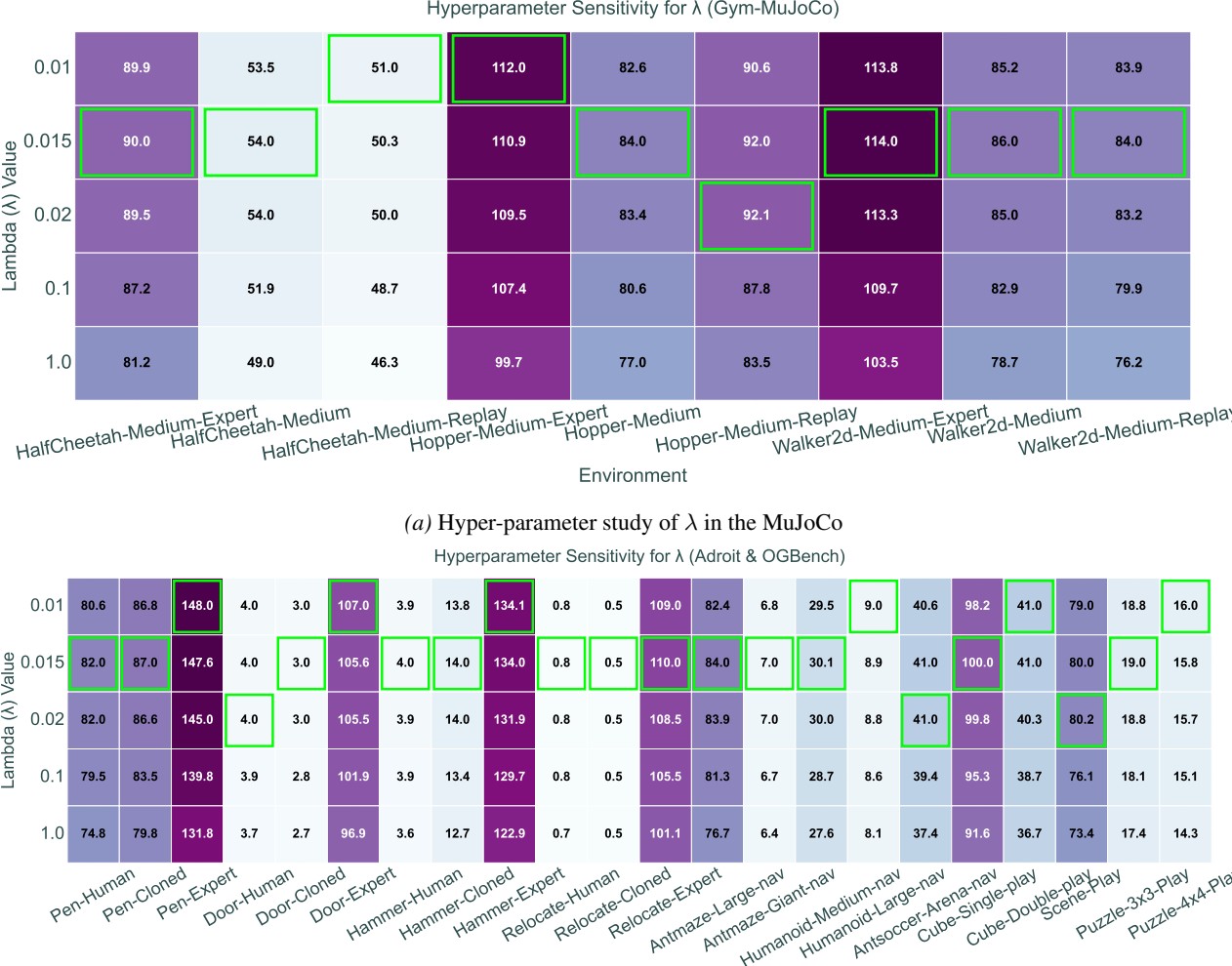

*(a)* Hyper-parameter study of $\lambda$ in the MuJoCo

*(b)* Hyper-parameter study of $\lambda$ in the Adroit and OGBench

*Figure 12.* Comprehensive hyper-parameter study of the PDE parameter $\lambda$ across all three benchmark suites. (a) shows results on Gym-MuJoCo tasks, while (b) shows results on Adroit and OGBench tasks.

---

**Algorithm 1** PDE-Regularized and Beta-Weighted Flow Matching for Offline RL

---

**Require:** Offline dataset $\mathcal{D} = \{(s, a, r, s')\}$, parameters $\theta$ for actor (velocity field $u_\theta$), parameters $\phi$ for critic $Q_\phi$, PDE regularization weight $\lambda$, Beta distribution $\pi_{\alpha,\alpha}$.

1: Initialize $\theta, \phi$ randomly.
2: **for** each training iteration **do**
3:    Sample minibatch $\{(s, a_1, r, s')\} \subset \mathcal{D}$.
4:    Sample noise actions $a_0 \sim \mathcal{N}(0, I)$.
5:    Sample timesteps $t \sim \pi_{\alpha,\alpha}$.
6:    Construct interpolants $a_t = (1 - t)a_0 + ta_1$.
7:    Compute predicted velocity $u_\theta(s, a_t, t)$.
8:    **Flow matching loss with Beta sampling:**

$$\mathcal{L}_\pi(\theta) = \mathbb{E}\left[w_t^\pi \|u_\theta(s, a_t, t) - (a_1 - a_0)\|^2\right], \quad w_t^\pi = \tfrac{t}{1-t}\,\pi(t).$$

9:    **Jacobian regularizer** (estimated with Hutchinson's trick)**:**

$$\mathcal{L}_{\mathrm{PDE}}(\theta) = \lambda\,\mathbb{E}\left[\|\nabla_a u_\theta(s, a_t, t)\|_F^2\right].$$

10:    **Critic loss:** Sample actions $\hat{a} \sim u_\theta(\cdot \mid s')$ and compute

$$\mathcal{L}_Q(\phi) = \mathbb{E}\left[\left(Q_\phi(s, a) - (r + \gamma\,Q_{\bar{\phi}}(s', \hat{a}))\right)^2\right].$$

11:    **Actor objective:**

$$\mathcal{L}_{\mathrm{actor}}(\theta) = \mathcal{L}_\pi(\theta) + \mathcal{L}_{\mathrm{PDE}}(\theta) - \mathbb{E}[Q_\phi(s, a_1')] + \mathcal{L}_{\mathrm{distill}}.$$

12:    Update actor parameters: $\theta \leftarrow \theta - \eta_\theta \nabla_\theta \mathcal{L}_{\mathrm{actor}}$.
13:    Update critic parameters: $\phi \leftarrow \phi - \eta_\phi \nabla_\phi \mathcal{L}_Q$.
14: **end for**

---

