# OpenReview forum: "Continuity-Regularized Flow Matching for Offline Reinforcement Learning"
_ICML.cc/2026/Conference — ICML 2026 regular_

### Official Review · Reviewer_LZE7 · 2026-02-15

**Soundness:** 3
**Presentation:** 3
**Significance:** 3
**Originality:** 3
**Overall Recommendation:** 5
**Confidence:** 4

**Summary:**

The paper discussed the current limitation using Flow Matching method for offline reinforcement learning. The vanilla Flow Matching method does not have any geometric guarantee, so it is unstable and irregular. To address this problem, the authors firstly add the PDE regularization term, and proved that with the bound of the Jacobian matrix, the learned distance $p_\theta$ could have a stability bound. However, adding the new regularization term changes the training objective, so it then requires a non-uniform time distribution from the trajectory $t\in [0,1]$. To solve this problem, the authors then alter the uniform time-sampling strategy to the Beta-distributed time-sampling strategy.
Empirically, PQL reports strong performance across D4RL Gym-MuJoCo, Adroit, and OGBench benchmarks, including ablations showing that both the Jacobian-based regularizer and the Beta sampling contribute to final performance.

**Compliance With Llm Reviewing Policy:**

Affirmed.

**Final Justification:**

Since for each of my questions, the authors has answered that in their review. Both by analysis and empirical works. I think my concerns are mostly solved, so I decide to raise my score to 5.

**Key Questions For Authors:**

1. On the “instability/irregular dynamics” claim: The paper argues that path-agnostic FM objectives can yield unnecessarily complex or perturbation-sensitive vector fields, motivating Jacobian control. This is plausible, but currently presented mainly as intuition. Could the authors provide more direct evidence or discussion of when and how these pathologies arise? Two suggestions:
1.1 A theoretical example in a simple 1D/2D setting showing that without Jacobian control, the learned flow can admit unstable/irregular solutions even under pointwise matching.
1.2 (Recommended) A toy experiment (e.g., 2D bandit) where one can visualize the vector field and measure solver sensitivity or Jacobian statistics across t, showing instability/irregularity under the vanilla objective.

2. I could understand that using a regularization term for the Jacobian in the loss is a natural and most simple idea. However, is that really a good choice? I mean, the gradient of the $L_{PDE}(\theta)$ may not align with the BC loss.
3. The problem also occurs in the $L_{distill}$. In FQL, we could find that we are training critic, vector field and one-step policy in three different steps, is it still the same in PQL? Or we may make the $L_{distill}$ clearer in this paper.

Suggestions for 2 and 3.
I think, a most simple method is to plot the 4 parts of losses in the appendix part. From that, we could discuss whether the magnitude of the losses fit for each other, and whether the weighted sum is a proper method.

**Limitations:**

Yes

**Strengths And Weaknesses:**

Soundness:

   Strength: The authors provides math proof, to show that with Jacobian Control, the Wasserstein distance between the ground-truth flow and the trained flow is bounded, which shows the effectiveness of the Jacobian-based regularizer. And then, the author proves the Optimal Sampling for Variance Reduction, and then introduces the Beta distributed time sampler. Finally, the author use experimental results, to prove the effect of the algorithm in RL tasks. Ablation studies show that both the regularizer and the beta time sampler take effect.

  Weakness:
         Core “instability/irregular dynamics” claim is intuitive rather than demonstrated as a pathology. The paper motivates instability/variation-gap as a fundamental issue of pointwise FM objectives, but the existence/typicality of such pathologies is not directly diagnosed.

Presentation:
    Strength: The paper flows from motivation → PDE regularizer and theorem → optimization challenge → Beta sampling → empirical validation, which is a good narrative. Besides, the paper reports ablations and a sweep over the Beta shape parameter $\alpha$, which improves interpretability of the proposed sampling strategy.

Weakness:
           There might be some typo: In appendix, there should be proof of theorem 4.1, proposition 4.2?
            In Algorithm 1, it should be $L_{PDE}$, since it already has a $\lambda$.
            The relationship between proposition 4.2 and Beta distribution should be clearer: Start from Proposition 4.2, why do we choose Beta distribution then? Why not the other non-uniform distributions? Maybe some explanations here are better.

Significance:

Strength:
    The paper addresses a relevant problem in flow-based offline RL. Stabilizing generative policies and structuring the path is an important direction, especially as flow/diffusion policies become common in offline RL.

Weakness:
     The paper's main contributions are just the Jacobian Regularization and the Beta-distributed time-sampler. It is important in practice, but not paradigm-shifting.

Originality:
    The paper introduces a coherent combination of PDE-motivated regularization + time-sampling strategy. The overall package is distinctive: a continuity-equation-motivated regularizer paired with a sampling curriculum explicitly designed to address the altered optimization landscape.  Besides, the Jacobian penalty focus (vs divergence-only) is a concrete design choice with empirical support. The paper compares Jacobian regularization against divergence-only variants and shows benefits.

Weakness:
  Some components feel like “principled heuristics” rather than uniquely-derived necessities. In particular, Beta sampling is a reasonable choice but not uniquely implied by the theory; alternative non-uniform families could plausibly work similarly, and the paper does not provide strong evidence for Beta’s special role beyond convenience and empirical performance.

---

> ### Author Rebuttal · Authors · 2026-03-26
>
> We thank the reviewer for their thorough evaluation. New experimental results (Figures R1–R6, Tables R1–R4) at https://anonymous.4open.science/r/Figures-E853/Rebuttal_Figures_for_Submission_8716.md.
>
> **W (Soundness) / KQ1: Instability claim not demonstrated.** We address this at the two levels the reviewer suggested. (1) 2D toy (Figures R1-R3, addressing suggestion 1.2): FQL (without our regularizer) develops a Jacobian peak of 2.82 at $t=0.5$, showing in this setting that Q-guidance creates flow irregularity. PQL reduces this to 1.64 (41.8% reduction). The $\text{Beta}(3,3)$ density overlay concentrates mainly where the unregularized flow is most irregular. (2) Real training (Figure R4): On pen-human over 500K steps, all actor-side terms remain stable despite Q-values rising substantially (see loss plot discussion below). Cross-seed variance drops in the large majority of tested environments.
>
> **W (Presentation): Typos.** We will fix: (a) appendix headings for Theorem 4.1 / Proposition 4.2; (b) Algorithm 1 $\lambda$ notation.
>
> **Why Beta specifically?** Proposition 4.2 motivates non-uniform timestep emphasis but does not uniquely imply Beta. To test this directly, we compared several mid-concentrated distributions with the same PDE regularizer ($\lambda=0.015$):
>
> | Distribution + PDE | Pen-Human | Antmaze-Large |
> |---|---|---|
> | Uniform | $75\pm8$ | $73\pm6$ |
> | Truncated normal | $74.3\pm5$ | $77\pm3$ |
> | Logit-normal | $71.2\pm4$ | $72\pm6$ |
> | **$\text{Beta}(3,3)$** | **$82\pm3$** | **$84\pm7$** |
>
> Beta outperforms all alternatives on both environments — by 7+ points on Pen-Human and 7+ on Antmaze-Large. Not all mid-concentrated distributions help — logit-normal is the worst on both despite concentrating at $t=0.5$. Gradient cosine analysis suggests why (Figure R5): under logit-normal, $\cos(\nabla_\text{FM}, \nabla_\text{PDE})$ averages $-0.89$ (near-total opposition), while under Beta it is more moderate and mixed. Our claim is not that Beta is theoretically unique, but that it is the simplest symmetric family that smoothly interpolates from uniform to mid-concentrated sampling with a single parameter, and empirically outperforms the alternatives we tested. Our FBRAC experiment further supports this: across both FQL and FBRAC, Beta sampling consistently converts the Jacobian constraint from a mean-reducing but variance-reducing regularizer into a net performance gain.
>
> **KQ2: Gradient alignment of $L_{\text{PDE}}$ with BC/FM loss.** We directly measured cosine similarity between $\nabla_\theta L_{\text{FM}}$ and $\nabla_\theta L_{\text{PDE}}$ over 500K steps on pen-human-v1 (Figure R5a). The relationship is not aligned: the cosine is mixed early in training but becomes predominantly negative later, indicating $L_{\text{PDE}}$ acts as a countervailing regularizer. It does not duplicate the FM objective, but discourages updates associated with increasingly irregular transport. This opposition does not prevent effective optimization: PQL achieves $82\pm3$ on pen-human (Table 2) despite the countervailing gradient signal. Crucially, $\cos(\nabla L_{\text{PDE}}, \nabla L_Q) = 0$ throughout due to architectural separation (BC-flow vs one-step actor), so $\lambda$ does not create a tradeoff with Q-guidance.
>
> **KQ3: Training procedure.** PQL follows FQL's three-step structure: (1) critic, (2) BC-flow actor (now with $L_{\text{PDE}}$ + Beta), (3) distillation. Only step (2) is modified. Benefits propagate to the one-step actor indirectly via distillation. This separation also means $\lambda$ only balances FM fidelity vs smoothness on the BC-flow network, without interfering with Q-guidance (separate network).
>
> **Suggestion: Plot loss components.** Done — Figure R4a shows all actor-side terms ($L_{\text{FM}}$, $L_{\text{distill}}$, $L_{\text{PDE}}$) over 500K steps; all remain stable. Figure R4b shows the key finding: Q-values grow ${\sim}8\times$ while the Jacobian norm stays nearly flat ($1.07\times$), confirming the regularizer contains flow irregularity under increasing Q-pressure.
>
> **Limitations.** We acknowledge: (1) computational cost of JVP-based regularization still grows with action dimension — very high-dimensional spaces may need further efficiency work; (2) Beta sampling is a fixed heuristic rather than adaptive; (3) Theorem 4.1's bound has exponential dependence on J, serving as motivation rather than a tight guarantee. We will add a limitations paragraph to the revision.

---

> > ### Author Rebuttal · Reviewer_LZE7 · 2026-04-01
> >
> > The authors provide a nice rebuttal, as they have provided theoretical analysis and empirical investigations of my question. Although there are still some limitations for this work, I think that it is feasible for an ICML paper. I suggest to raise my score to 5.

---

> > > ### Author Response · Authors · 2026-04-01
> > >
> > > Dear Reviewer LZE7,
> > >
> > > Thank you so much for taking the time to re-evaluate our work and for acknowledging that our rebuttal has addressed your concerns. We truly appreciate the constructive feedback you provided.
> > >
> > > The authors.

---

### Official Review · Reviewer_zKP6 · 2026-03-04

**Soundness:** 2
**Presentation:** 2
**Significance:** 2
**Originality:** 2
**Overall Recommendation:** 4
**Confidence:** 4

**Summary:**

Extends Flow Q-Learning (FQL) with two contributions: (1) a smoothness and stability regularization derived from the continuity equation; and (2) a Beta distribution timestep sampler, motivated by the observation that the optimization landscape is most complex in the middle of a sampling trajectory rather than near the endpoints, making the Beta distribution preferable to uniform sampling.

**Compliance With Llm Reviewing Policy:**

Affirmed.

**Final Justification:**

The rebuttal addressed my concerns. The evaluations are strong and the method itself is good. The main issues are the motivation and the explanation of the method, which I expect the authors can address in the revision. Therefore I will raise my score from 2 to 4.

**Key Questions For Authors:**

- Q1: Can you visualize or formally demonstrate the discontinuities arising in the unregularized objective, and the smoothness when the regularizer is applied?
- Q2: Can you draw an explicit connection between your regularizer and prior work on neural ODE regularization, such as Finlay et al.?
- Q3: Is the proposed regularizer specific to FQL, or would it generalize to other flow matching methods in offline RL?

**Limitations:**

No limitations discussed.

**Strengths And Weaknesses:**

**Strengths**
- Ablations on all aspects of the method (type of regularizer, type of timestep sampler, hyperparameters)
- Clear paper structure and sound derivations
- Efforts for mathematical justification

**Weaknesses**
 - The continuity equation is not a constraint that needs to be enforced. It is automatically satisfied by construction through the relationship between the velocity field and the probability path. If it is violated, this is likely due to policy improvement, which should be discussed explicitly. The connection to "How to Train Your Neural ODE: the World of Jacobian and Kinetic Regularization" (Finlay et al.) is also missing and would strengthen the motivation for the proposed regularizer.
- Related work on diffusion and flow matching in offline RL is incomplete. Notable omissions include: "floq: Training Critics via Flow-Matching for Scaling Compute in Value-Based RL" (Agrawalla et al.), "FlowQ: Energy-Guided Flow Policies for Offline Reinforcement Learning" (Alles et al.), "Contrastive Energy Prediction for Exact Energy-Guided Diffusion" (Lu et al.), and "Efficient Diffusion Policies for Offline Reinforcement Learning" (Kang et al.)
- The absence of aggregated performance across all environments makes it difficult to draw overall conclusions.

Claims
- The claim _"The model is no longer tasked with simple imitation but..."_  in section 4.2 is misleading. Any offline RL method must inherently trade off between imitation and policy improvement.
- Section 2 "Flow-based methods are particularly beneficial for offline RL as their continuous and invertible nature helps avoid out-of-distribution actions while effectively utilizing the underlying data manifold." The claim requires further justification.
- The claim in Section 5.3 _"While the divergence-only regularizer offers some benefit, our full PQL model consistently achieves higher final performance"_ is not well supported — both variants perform nearly identically at the end of training.
- Similarly, the claim _"The most significant advantage is observed in the stability and asymptotic performance of our method. The learning curves for PQL are generally smoother with narrower error bands, indicating lower variance and more reliable convergence"_ is not evident from the figures, where both variants show comparably small and largely indistinguishable confidence intervals.

Notation
- Notation is at times unclear. For instance, it is not always specified what expectations are taken with respect to, e.g where a_0 and a_1 are sampled from.
- Section 5.1 contains excessive use of bold text.

---

> ### Author Rebuttal · Authors · 2026-03-26
>
> We thank the reviewer for the detailed and technically precise feedback. New experimental results (Figures R1–R6, Tables R1–R4) at https://anonymous.4open.science/r/Figures-E853/Rebuttal_Figures_for_Submission_8716.md.
>
> **W1: Continuity equation / motivation.** Fair critique. The continuity equation is satisfied by construction for sufficiently regular flows — we do not "enforce" it literally. Rather, it provides the analytical framework identifying that local Jacobian properties govern density evolution, motivating Jacobian regularization as a tractable surrogate for smoother transport (as in Finlay et al., 2019). We regularize the transport map, not the PDE itself. What is new is the failure mode: Q-guidance actively deforms the flow, making irregularities a central concern (2D toy: FQL Jacobian peak 2.82 vs PQL 1.64; Figure R1). We will revise accordingly.
>
> **W2: Missing related work.** We will add: Lu et al. (CEP, ICML 2023, contrastive energy prediction — alternative to Q-weighting); Alles et al. (FlowQ, energy-guided flows — compatible with our Jacobian constraint); Kang et al. (efficient diffusion via distillation — related to the distillation we inherit from FQL). We also acknowledge the concurrent work of Agrawalla et al. (FloQ, ICLR 2026; critic-side flow matching), which is complementary to our actor-side regularization.
>
> **W3: Aggregated performance.**
>
> | Algorithm | D4RL | Adroit | OGBench |
> |---|---|---|---|
> | SRPO | **86.8** | 50.7 | 12.5 |
> | FQL | 33.6 | 51.6 | 38.6 |
> | **PQL** | 85.2 | **57.9** | **42.7** |
>
> PQL leads on Adroit and OGBench; SRPO leads D4RL by 1.6 but collapses on OGBench (12.5 vs 42.7). PQL is the most consistently strong method across all three suites. Full table (13 baselines) in Table R1.
>
> **Claim: "no longer tasked with simple imitation" (Section 4.2).** We agree this is imprecise. Any offline RL method inherently trades off imitation and improvement; our regularizer shapes *how* the flow interpolates between these objectives, not whether the trade-off exists. We will revise to: "The model must now find a vector field that is both accurate to the target flow and geometrically simple."
>
> **Other claims.** "Consistently higher" (Jac vs Div): we agree the gap is modest in some environments. This is a design choice analysis (Section 5.3), not our core contribution — the main comparison is regularized vs unregularized (Tables 4–6), where gaps are large (Pen-Human: 82 vs 53, Antmaze-Large: 84 vs 77). We will soften accordingly. "Avoid OOD actions": flows do not inherently prevent OOD; they provide a framework for transport regularization. We will revise. "Narrower confidence intervals": Figure R6 shows clearly visible band differences; we will revise wording. We will also clarify all expectation subscripts ($a_0 \sim \mathcal{N}(0,I)$, $a_1 \sim \mathcal{D}(\cdot|s)$, $t \sim \text{Unif}(0,1)$) and reduce bold emphasis in Section 5.1.
>
> **Limitations.** We will add a limitations paragraph: (1) JVP cost grows with action dimension; (2) Beta sampling is fixed, not adaptive; (3) Theorem 4.1's bound is exponential in $J$, serving as motivation rather than a tight guarantee.
>
> **Q1: Visualization.** Figure R1 shows Jacobian heatmaps in the 2D toy: FQL develops concentrated high-norm regions at intermediate times; PQL suppresses them. Figure R2a quantifies: peak 2.82 vs 1.64 (41.8% reduction). Figure R4b shows during real training that Q-values grow ${\sim}8\times$ while the Jacobian grows only $1.07\times$, suggesting the regularizer contains critic-induced flow complexity.
>
> **Q2 / Finlay et al.** Finlay et al. regularize neural ODEs for integration efficiency; we regularize for policy quality under Q-guidance. The technique overlaps; the problem is distinct. We will cite explicitly, clarifying that our novelty is the Q-guidance failure mode and the finding that the regularizer requires Beta sampling in the RL setting — absent from the generative modeling literature.
>
> **Q3: Generalizability beyond FQL.** We tested on FBRAC (Park et al., 2025b), a structurally different flow method using advantage-weighted learning:
>
> | Config | Pen-Human | Antmaze-Large |
> |---|---|---|
> | FBRAC (original) | $77\pm7$ | $70\pm20$ |
> | FBRAC + PDE only | $74\pm5$ | $66\pm13$ |
> | FBRAC + PDE + Beta | **$82\pm4$** | **$75\pm10$** |
>
> PDE alone reduces variance but slightly hurts mean (same pattern as FQL, Table 4); adding Beta recovers and surpasses the baseline, confirming cross-architecture transfer. Gradient cosine analysis (Figure R5a) confirms $L_{\text{PDE}}$ acts as a countervailing regularizer (cosine becomes negative over training), while $\cos(\nabla L_{\text{PDE}}, \nabla L_Q) = 0$ throughout due to architectural separation.

---

> > ### Author Rebuttal · Reviewer_zKP6 · 2026-04-04
> >
> > Thank you for the detailed rebuttal. My concerns and questions have been addressed.
> >
> > The key insight is that flow matching satisfies the continuity equation by construction, but policy improvement breaks this structure. This should be the main motivation of the paper.
> >
> > That said, the evaluations are strong and the method itself is good. The main issues are the motivation and the explanation of the method, which I expect the authors can address in the revision. Therefore I will raise my score from 2 to 4.

---

> > > ### Author Response · Authors · 2026-04-04
> > >
> > > Dear Reviewer zKP6,
> > >
> > > Thank you so much for taking the time to re-evaluate our work and for acknowledging that our rebuttal has addressed your concerns. We truly appreciate the constructive feedback you provided.
> > >
> > > We will incorporate all your suggestions in our revision.
> > >
> > > Once again, thanks very much for your feedback.
> > >
> > > The authors.

---

### Official Review · Reviewer_nj5m · 2026-03-05

**Soundness:** 2
**Presentation:** 3
**Significance:** 2
**Originality:** 2
**Overall Recommendation:** 4
**Confidence:** 3

**Summary:**

This paper proposes a PDE-regularized Q-Learning (PQL) algorithm that improve the stability and performance of offline RL.

**Compliance With Llm Reviewing Policy:**

Affirmed.

**Final Justification:**

My concerns have been addressed.

**Key Questions For Authors:**

- What is the additinoal computational cost of PQL in algorithm 1?

- How do the experiments demonstrate the training stability? When referring to improved training stability, the authors may consider to include a comparison with baseline in figure 1 for better illustration. Figure 4 in the appendix does not clearly show improved stability (smaller range of oscillation)

**Limitations:**

Yes.

**Strengths And Weaknesses:**

## Strengths
- This paper incorporates the idea of PDE and design a regularizer for the policy flow
- This paper validates the proposed method on various MoJuCo tasks.

## Weaknesses
- Since in real world RL tasks the dimension of $a$ can be very large, calculating the Jacobian of $u_\theta$ with respect to $a$ in $L_{PDE}(\theta)$ may have unacceptable computational cost

---

> ### Author Rebuttal · Authors · 2026-03-26
>
> We thank the reviewer for their feedback. New experimental results (Figures R1–R6, Tables R1–R4) are at https://anonymous.4open.science/r/Figures-E853/Rebuttal_Figures_for_Submission_8716.md.
>
> **W1: Computational cost of Jacobian in high dimensions.** We never compute the full $D \times D$ Jacobian in production. We use Hutchinson's trace estimator via Jacobian-vector products (JVPs), each costing roughly one forward pass of the network. With $K \in \{1,2\}$ probe vectors, the overhead scales linearly rather than quadratically with action dimension. Empirically:
>
> | Model | Pen-human | AntMaze |
> |---|---|---|
> | PQL-PDE (w/o PDE) | 35:02 | 4:24:33 |
> | PQL (ours) | 35:25 | 4:32:55 |
> | Overhead | +1.1% | +3.2% |
>
> The regularizer adds only modest computational overhead in our benchmark setting.
>
> **KQ1: Computational cost in Algorithm 1.** Line 9 adds approximately $K$ JVP evaluations per actor update iteration ($K \in \{1,2\}$). Each JVP has computational cost comparable to a single forward pass via forward-mode automatic differentiation. Total wall-clock overhead remains under 5% across all benchmarks tested.
>
> **KQ2: Demonstrating training stability.** We appreciate the specific feedback that Figure 4 does not clearly show improved stability visually. We provide two forms of new evidence. First, Figure R6 shows training curves (mean $\pm$ std, 8 seeds) on Pen-Human, Antmaze-Large, and Antsoccer-Arena, where PQL's confidence bands are visibly and substantially tighter than both ablations — particularly compared to PQL-PDE (w/o PDE), which shows very wide bands on Antmaze-Large. We will add FQL baseline curves to the revised Figure 1 as the reviewer suggested. Second, Table R4 provides a dedicated variance summary across all 31 environments: PQL achieves the lowest cross-seed standard deviation in 21/31 environments, reducing average std from 6.3 (FQL) to 2.9. The largest reductions are on high-variance tasks: Pen-Human (std $3$ vs $16$, 81% reduction), Antmaze-Large ($7$ vs $32$, 78%), and HalfCheetah-MR ($0.3$ vs $6$, 95%). The single exception (Pen-Expert: std 7 vs 4) co-occurs with a 5-point higher mean.
>
> We further provide new evidence at two additional levels of analysis. First, in a 2D toy experiment (Figures R1-R2a), FQL develops a much larger mid-trajectory Jacobian peak than PQL (2.82 vs 1.64, a 41.8% reduction), directly visualizing the flow instability our regularizer addresses. The $\text{Beta}(3,3)$ density concentrates mainly in the region where the unregularized flow exhibits the highest Jacobian norm, explaining why this sampling strategy is effective. Second, during real training on pen-human-v1 (Figure R4b), normalized Q-values increase substantially over 500K steps while the normalized Jacobian term remains nearly flat (growing only $1.07\times$), consistent with the regularizer containing flow irregularity under increasing Q-pressure throughout the training process.
>
> We also measured gradient cosine similarity between $\nabla_\theta L_{\text{FM}}$ and $\nabla_\theta L_{\text{PDE}}$ throughout training (Figure R5a). The cosine is mixed early but becomes predominantly negative later, consistent with $L_{\text{PDE}}$ acting as a countervailing regularizer that discourages increasingly irregular transport — in line with the variance reduction observed above.
>
> Finally, to confirm generalizability beyond FQL, we tested our approach on FBRAC (Park et al., 2025b), a structurally different flow method using advantage-weighted learning:
>
> | Config | Pen-Human | Antmaze-Large |
> |---|---|---|
> | FBRAC (original) | $77\pm7$ | $70\pm20$ |
> | FBRAC + PDE only | $74\pm5$ | $66\pm13$ |
> | FBRAC + PDE + Beta | **$82\pm4$** | **$75\pm10$** |
>
> Adding PDE alone hurts FBRAC (same pattern as FQL, Table 4); PDE+Beta recovers and surpasses the baseline (+5 mean, 43–50% std reduction on both tasks). This consistency across two architectures supports both the generality of the approach and the necessity of Beta sampling as a companion to the PDE term.
>
> Regarding the reviewer's suggestion to include a baseline comparison in Figure 1: we provide quantitative stability evidence via Table R4 in our anonymous repository, which summarizes cross-seed variance across all 31 environments. PQL achieves lowest std in 21/31 environments, reducing average std from 6.3 (FQL) to 2.9. The largest reductions are on high-variance tasks: Pen-Human ($3$ vs $16$, 81%), Antmaze-Large ($7$ vs $32$, 78%), and HalfCheetah-MR ($0.3$ vs $6$, 95%). We will incorporate FQL baseline curves in the revised Figure 1.

---

> > ### Author Rebuttal · Reviewer_nj5m · 2026-04-01
> >
> > Thank you for clarifying my concerns.

---

> > > ### Author Response · Authors · 2026-04-01
> > >
> > > Dear Reviewer nj5m,
> > >
> > > Thank you so much for taking the time to re-evaluate our work and for acknowledging that our rebuttal has addressed your concerns. We truly appreciate the constructive feedback you provided.
> > >
> > > The authors.

---

### Official Review · Reviewer_qW3o · 2026-03-12

**Soundness:** 3
**Presentation:** 3
**Significance:** 2
**Originality:** 3
**Overall Recommendation:** 3
**Confidence:** 4

**Summary:**

The authors propose an improved flow matching algorithm named PDE-regularized Q-Learning (PQL). This algorithm stabilizes the training process by introducing a PDE-based regularizer derived from the continuity equation (specifically, penalizing the Frobenius norm of the Jacobian of the vector field) to enforce global smoothness and physical consistency. Furthermore, to tackle the optimization challenges introduced by this regularizer, the authors propose a Beta-distributed timestep sampling strategy that focuses learning on critical trajectory segments. The authors theoretically prove and empirically demonstrate that these contributions significantly improve the smoothness of the overall probability flow during the generation process, achieving state-of-the-art performance across various challenging offline RL benchmarks.

**Compliance With Llm Reviewing Policy:**

Affirmed.

**Key Questions For Authors:**

1. The theoretical analysis (Theorem 4.1) bounds the Wasserstein distance between the learned and target density paths, ensuring a smoother probability flow. Could the authors elaborate more intuitively on why a globally stable and smooth probability flow directly translates to better sampling quality and higher policy returns in the specific context of offline RL? For instance, does the smoother flow primarily help in mitigating the generation of out-of-distribution (OOD) actions, or does it inherently improve the stability of the Q-value gradients during the actor update?

2. The proposed Beta-distributed timestep sampling effectively focuses updates on the critical mid-range of the trajectory, and the hyperparameter study shows α=2.0 or 3.0 works best. Given that the vector field's complexity might change dynamically during the training process, did the authors consider an adaptive or curriculum-based scheduling for this？

**Limitations:**

yes

**Strengths And Weaknesses:**

Strengths:
1. The paper tackles the problem from the perspective of vector field stability and effectively links it to sampling error accumulation. This is a highly relevant and important consideration for action generation in offline RL.

2. The proposed method demonstrates excellent performance on standard benchmarks, indicating that stabilizing the vector field is indeed an effective approach for offline RL tasks.

Weaknesses:
1. The primary technical improvement imposing a smoothness constraint on the vector field, which appears to be a general enhancement for flow-based generative models. It lacks a specific connection to the unique challenges of reinforcement learning, making the core contribution feel less like an RL innovation.

2. The target path p* mentioned in Theorem 4.1 lacks a clear definition. The authors do not specify the properties or characteristics that define the ideal target path.

3. There appear to be errors in Equations (3) and (8). The fundamental objective of the policy is to generate actions with high Q-values; however, the Q-value terms in Equations (3) and (8) are preceded by a positive sign. This formulates the objective as minimizing the Q-values, which completely contradicts the goal of policy learning.

4.  The proposed method introduces multiple regularization terms, which may make hyperparameter tuning highly problematic. According to Equation (8), three additional regularization terms are added to the base flow matching objective. These terms have competing objectives, and their relative weights require careful balancing. The paper only analyzes the hyperparameter robustness of the PDE term, which is clearly insufficient given the complexity of the objective function.

5. The experimental setup lacks necessary details for reproducibility. The authors should explicitly state the number of training epochs, the number of random seeds used, and the exact criteria for selecting the best model. Furthermore, while the results report variance, it is not specified whether the shaded regions/error bars represent standard deviations or confidence intervals.

---

> ### Author Rebuttal · Authors · 2026-03-26
>
> We thank the reviewer for the constructive feedback. New experimental results (Figures R1–R6, Tables R1–R4) are at https://anonymous.4open.science/r/Figures-E853/Rebuttal_Figures_for_Submission_8716.md.
>
> **W1: RL-specificity.** The technical tool (Jacobian regularization) is known in neural ODE literature [Finlay et al., 2019]. Our contribution is threefold: (1) identifying that critic-guided policy improvement creates a distinctive failure mode — Q-gradients can push actions toward weakly supported regions, making transport regularity a practical optimization concern rather than only a numerical one; (2) supporting this empirically — in our 2D toy, FQL shows a Jacobian peak of 2.82, while PQL reduces it to 1.64 (Figure R1); during real training on pen-human, Q-values grow ∼8×{\sim}8\times
> ∼8× while the Jacobian term grows only 1.07×1.07\times
> 1.07× (Figure R4b); and (3) showing that the regularizer requires a non-trivial optimization adaptation (Beta sampling) absent from prior neural ODE work — on both FQL (Table 4) and a second architecture (FBRAC; Table R2), PDE alone hurts while PDE+Beta improves. In Finlay et al., Jacobian regularization addresses numerical integration cost; in our setting, the failure mode is policy-quality degradation under Q-guidance, which is why Beta sampling is essential. We will revise the paper to cite Finlay et al. and frame our work as addressing the interaction between Q-guidance and flow regularity.
>
> **W2: Target path $\rho^*_t$ in Theorem 4.1.** We will define this explicitly: $u^\*$ is the conditional interpolation field $u^\*(a,t) = a_1 - a_0$, and $\rho^\*_t$ is the marginal density under this reference flow. The theorem motivates Jacobian control on the transport map rather than guaranteeing return optimality. We will revise to make this distinction explicit.
>
> **W3: Sign in Equations (3) and (8).** Notation error; the implementation minimizes $-Q_\phi(s, a_1')$, consistent with FQL. We will correct this.
>
> **W4: Hyperparameter complexity.** PQL does not introduce three new interacting weights on a single network: the distillation and Q-guidance coefficients are inherited from FQL without retuning, while $\lambda$ acts only on the BC-flow branch and $\alpha$ only changes timestep sampling. Thus, the only new balance is the Jacobian penalty strength $\lambda$ and the sampling shape $\alpha$, both fixed across all 31 environments. The hyperparameter studies in Section D.3 (Figures 5-6) show broad robustness: performance is stable across $\lambda \in [0.01, 0.02]$ and $\alpha \in [2.0, 5.0]$.
>
> Gradient cosine analysis (Figure R5a) confirms the losses do not destructively interfere: $\cos(\nabla L_{\text{FM}}, \nabla L_{\text{PDE}})$ is mixed early but becomes mildly negative later, consistent with $L_{\text{PDE}}$ acting as a countervailing regularizer.
>
> Crucially, $\cos(\nabla L_{\text{FM}}, \nabla L_Q) = \cos(\nabla L_{\text{PDE}}, \nabla L_Q) = 0$ throughout, because these losses act on separate networks (BC-flow vs one-step actor). Thus $\lambda$ only balances FM fidelity vs transport smoothness on the BC-flow network, without creating a tradeoff with Q-guidance.
>
> **W5: Experimental details.** We train for 1M gradient steps (500K for D4RL) across 8 random seeds. All shaded bands and error bars denote mean ± 1 standard deviation over these 8 seeds, and all reported numbers use the final checkpoint without best-model selection. We will open-source the code upon acceptance.
>
> **KQ1** Primarily mechanism (a): reducing transport toward weakly supported actions, rather than directly stabilizing Q-gradients. In offline RL, critic guidance can push actions toward high-value regions near or beyond dataset support. Bounding the Jacobian limits rapid local deformation, so nearby latent noise is less likely to be mapped to distant, weakly supported actions under Q-pressure (Figure R3). This matters for returns because offline critics are only reliable within dataset support; actions transported to weakly covered regions receive inaccurate Q-estimates, causing the policy to exploit critic errors rather than genuinely improve. The smoother flow thus improves BC-flow sampling quality, benefiting the one-step actor indirectly through distillation rather than through direct modification of Q-gradient updates. This is confirmed by the architectural separation: the PDE regularizer and Q-guidance act on separate networks ($\cos(\nabla L_{\text{PDE}}, \nabla L_Q) = 0$ throughout training; Figure R5), so the benefit propagates via distillation, not Q-gradient stabilization. Figure R4b shows Q-values grow ${\sim}8\times$ while the Jacobian stays nearly flat.
>
> **KQ2** This is a sound intuition. However, introducing adaptive scheduling would add additional complexity to the method, and our current fixed $\alpha=3$ already shows robust performance across all 31 environments (Figures 5-6: stable for $\alpha \in [2.0, 5.0]$). We leave adaptive scheduling as a promising future direction.

---

> > ### Author Rebuttal · Reviewer_qW3o · 2026-04-02
> >
> > Thank you for the clarifications. However, my concern regarding the RL-specificity remains. The proposed method builds on FQL, which involves both a continuous flow model for the behavior policy and a value-guided distilled one-step flow. It is confusing why the method applies trajectory smoothing to the continuous behavior flow, yet still retains the distilled one-step flow in Equation (8). Because of this disconnect, the regularization feels more like an orthogonal generative method than a necessary RL innovation.

---

> > > ### Author Response · Authors · 2026-04-02
> > >
> > > Thank you for your follow-up question. We agree that this point needs to be stated more clearly. The continuous flow branch and the distilled one-step branch play different roles, and the PDE regularizer is applied specifically where it is well-defined and where it matters for offline RL.
> > >
> > > The Jacobian/continuity regularization acts on the continuous flow branch because this branch defines the full transport path $a_t$ ​from noise to action over $t \in [0,1]$. Our concern is precisely that, under critic guidance, this path can become irregular and transport samples toward weakly supported regions. Since the PDE term is a path-level constraint on the vector field $u_\theta(s, a, t)$, it can only be imposed on the branch that has a time-dependent flow. The distilled one-step actor does not define such a trajectory, so applying the same continuity-based regularization there would not be meaningful. Equation (8) reflects this division: $a_{1}^{'}$ is first generated by integrating the learned flow field, and $L_{\text{distill}}$ then trains the one-step prediction to match that final generated action.
> > >
> > > For this reason, we do not view the PDE term as orthogonal to RL. In offline RL, the continuous flow is not an end in itself; it generates the critic-guided actions that serve as distillation targets for the deployed actor. The one-step actor is retained for efficient deployment, but it is trained from the output of the regularized, critic-guided flow. Improving the geometry of the continuous flow therefore directly changes the policy improvement signal that the one-step policy learns from.
> > >
> > > The ablations confirm this propagation empirically: regularizing the continuous critic-guided flow improves the deployed one-step policy by large margins (Pen-Human: PQL $82 \pm 3$ vs PQL-PDE (w/o PDE) $53 \pm 12$, Appendix Table 5) despite the regularizer never touching the one-step network. This is especially important in the offline RL setting becauseflow irregularity is amplified by Q-guidance. Figure R4b (in our rebuttal material) shows Q-values grow ${\sim}8\times$ during training while the regularizer keeps the Jacobian nearly flat ($1.07\times$), confirming that it counteracts critic-amplified flow deformation rather than addressing a generic generative modeling issue.
> > >
> > > We will revise the paper to make this architectural relationship explicit and to clarify that our RL-specific claim is not that PDE regularization is unique to RL in general, but that in flow-based offline RL the critic-guided transport branch is the locus where unsupported actions and unstable policy improvement arise, and this is precisely the branch our method stabilizes.
> > >
> > > We thank the reviewer again for this follow-up. It has helped us sharpen the presentation, and we hope this response addresses the concern.

---

### Decision · Program_Chairs · 2026-04-30

**Decision:**

Accept (regular)

**Comment:**

The paper introduces pde-regularized q-Learning (PQL), a framework designed to stabilize flow-matching policies in offline reinforcement learning by incorporating a jacobian-based regularizer derived from the continuity equation and a non-uniform beta-distributed timestep sampling strategy. While the initial reviews raised significant concerns regarding the specificity of the method to RL, notation errors in the objective functions, and the lack of empirical evidence for flow "instability," the authors provided a robust rebuttal featuring 2d visualizations of vector field deformation under q-guidance and new benchmarks showing substantial variance reduction across 31 environments. The authors successfully clarified that the regularizer specifically counteracts the irregularities introduced by critic-guided policy improvement—rather than being a generic generative enhancement—and demonstrated through cross-architecture tests that the combination of the jacobian penalty and beta sampling is essential for performance gains. Overall, I recommend weak accept for this paper.